# Technical note: High-resolution analyses of concentrations and sizes of refractory black carbon particles deposited on northwest Greenland over the past 350 years – Part 1. Continuous flow analysis of the SIGMA-D ice core using a Wide-Range Single-Particle Soot Photometer and a high-efficiency nebulizer

Kumiko Goto-Azuma[1,2], Remi Dallmayr[1,a], Yoshimi Ogawa-Tsukagawa[1], Nobuhiro Moteki[3], Tatsuhiro Mori[4], Sho Ohata[5], Yutaka Kondo[1], Makoto Koike[6], Motohiro Hirabayashi[1], Jun Ogata[1], Kyotaro Kitamura[1], Kenji Kawamura[1,2], Koji Fujita[5], Sumito Matoba[7], Naoko Nagatsuka[1,c], Akane Tsushima[1,b], Kaori Fukuda[1], and Teruo Aoki[1]

[1]National Institute of Polar Research, Tachikawa, Tokyo, 190-8518, Japan

[2]SOKENDAI, Shonan Village, Hayama, Kanagawa, 240-0193, Japan

[3]Tokyo Metropolitan University, Hachioji, Tokyo, 192-0397, Japan

[4]Keio University, Yokohama, Kanagawa, 223-8521, Japan

[5]Nagoya University, Nagoya, 464-8601, Japan

[6]University of Tokyo, Bunkyo-ku, 113-0033, Japan

[7]Hokkaido University, Sapporo, 060-0819, Japan

[a]Now at Alfred Wegener Institute for Polar and Marine Research, Bremerhaven, Germany

[b]Now at Meteorological Research Institute, Tsukuba, Ibaraki, 305-0052, Japan

[c]Now at Japan Agency for Marine-Earth Science and Technology, Yokosuka, Kanagawa, 237-0061, Japan

*Correspondence to*: Kumiko Goto-Azuma (kumiko@nipr.ac.jp)

**Abstract.** Ice cores can provide long-term records of refractory black carbon (rBC), an important aerosol species closely linked to the climate and environment. However, previous studies of ice cores only analysed rBC particles with diameter of <500nm, which could have led to underestimation of rBC mass concentrations. Information on the size distribution of rBC particles is very limited, and there are no Arctic ice core records of the temporal variation in rBC size distribution. In this study, we applied a recently developed improved technique to analyse the rBC concentration in an ice core drilled at the SIGMA-D site in northwest Greenland. The improved technique, which uses a modified Single-Particle Soot Photometer and a high-efficiency nebulizer, widens the measurable range of rBC particle size. For high-resolution continuous analyses of ice cores, we developed a continuous flow analysis (CFA) system. Coupling of the improved rBC measurement technique with the CFA system allows accurate high-resolution measurements of the size distribution and concentration of rBC particles with diameter between 70 nm and 4 μm, with minimal particle losses. Using this technique, we reconstructed the size distributions and the

number and mass concentrations of rBC particles during the past 350 years. On the basis of the size distributions, we assessed the underestimation of rBC mass concentrations measured using the conventional SP2s. For the period 2003–2013, the underestimation of the average mass concentration would have been 12%–31% for the SIGMA-D core.

## 1 Introduction

Black carbon (BC), which is emitted from both anthropogenic and natural sources (e.g., fossil fuel combustion and biomass burning), can affect Earth's radiation budget by absorbing sunlight and reducing the albedo of snow and ice surfaces (e.g., Bond et al., 2013; Mori et al., 2019; Matsui et al., 2022; Moteki, 2023 and references therein). Particles of BC can also affect cloud microphysical processes by acting as cloud condensation nuclei (CCN) or ice nucleating particles (e.g., Bond et al., 2013; AMAP, 2021), thereby indirectly affecting the radiation budget. Over the past half-century, the Arctic has warmed at a rate four times faster than that of the global average (Rantanen et al., 2022), leading to drastic changes such as sea ice retreat, enhanced losses of glacier mass, and ecosystem changes. It is therefore important to evaluate the effects of BC on the radiation budget in the Arctic. Freshly emitted BC particles are initially hydrophobic, but gradually become coated with other aerosol species, transforming into internally mixed hydrophilic particles during transport (*e.g.* Mori et al., 2017; Matsui, 2017; Matsui and Mahowald, 2017). These hydrophilic BC particles can be activated as CCN, depending on their size and mixing state, and are eventually deposited on the earth's surface via precipitation. The size distribution of BC particles influences not only their ability to act as CCN but also their transport and deposition processes, thereby controlling the temporal and spatial variability of BC concentrations. In addition, size distribution affects the light absorption properties of BC particles. Therefore, the size distribution as well as concentrations of BC particles is a key parameter for understanding the impacts of BC on Earth's radiation budget. Data acquired since the pre-industrial period are particularly valuable because we cannot fully understand the anthropogenic effects without characterizing BC in a pristine environment. However, no direct measurements of the size distributions and concentrations of BC particles were performed prior to the past few decades despite numerous studies based on observations and aerosol/climate models (e.g., Bond et al., 2013 and references therein).

Ice cores can provide long-term records of BC deposition. Following development of the Single-Particle Soot Photometer (SP2; Droplet Measurement Technologies, USA) (Stephens et al., 2003; Baumgardner et al., 2004), it has been possible to measure refractory black carbon (rBC), the terminology used for incandesce-based BC measurements (Petzold et al., 2013; Lim et al., 2014), even in Arctic and Antarctic ice cores that contain very low concentrations of rBC particles

(McConnell et al., 2007; Zdanowicz et al., 2018; Osmont et al., 2018; Zennaro et al., 2014; Bisiaux et al., 2012a, b; Arienzo et al., 2019). Moreover, attachment of a coupled SP2 and nebuliser system to a continuous flow analysis (CFA) system allowed continuous and high temporal-resolution analyses of rBC in ice cores drilled at a site with little summer melting (McConnell et al., 2007; Lim et al., 2017; Bisiaux et al., 2012a, 2012b; Arienzo et al., 2017). Many previous SP2 analyses of rBC in ice cores, regardless of whether they used a CFA system, adopted the U5000AT ultrasonic nebuliser (Teledyne CETAC, USA) system (or a similar ultrasonic nebuliser) to aerosolize rBC particles in melted ice core samples before their introduction to the SP2 (McConnell et al., 2007; Zennaro et al., 2014; Zdanowicz et al., 2018; Du et al., 2020; Kaspari et al., 2011; Wang et al., 2015; Bisiaux et al., 2012a, 2012b: ;rienzo et al., 2017). Owing to the complex and temporally variable size dependence of the extraction efficiency of the U5000AT ultrasonic nebuliser system (Schwarz et al., 2012; Wendl et al., 2014; Ohata et al., 2013; Mori et al., 2016), large uncertainties are associated with the derived size distributions and concentrations. Obtaining accurate estimation of the size distribution of rBC particles on a routine basis is not easy using the U5000AT nebulizer system. While Kaspari et al. (2011) reported mass size distributions of rBC in two samples from a Mt. Everest ice core using the U5000AT nebulizer system, long-term ice core records of the size distribution of rBC particles obtained using this type of nebulizer system have not been reported. On the contrary, Wendl et al. (2014) demonstrated size-independent extraction efficiency (<15% variability) of the APEX Q jet nebulizer system (High-Sensitivity Sample Introduction System, Elemental Scientific Inc., USA) for rBC particles in the 100-1000 nm diameter rage. Lim et al. (2014) also reported size-independent extraction efficiency (<10% variability) of the APEX Q nebulizer system for rBC particles with diameters between 60 and 500 nm. As a result, recently, the APEX-Q nebulizer system is becoming the standard within the ice core community. Using an APEX Q nebulizer system and an SP2 attached to a CFA system, Lim et al. (2017) analysed ice cores from Mt. Elbrus (western Caucasus Mountains) and reported temporal variability in the size and concentration of rBC particles with diameters between 70 and 620 nm during 1825–2013. However, to date, no BC size distribution data from Arctic ice cores have been published.

Snow and hence ice cores could contain much larger BC particles than those typically observed in the atmosphere (Schwarz et al., 2012, 2015). The particle size range typically measurable by an off-the-shelf SP2 is from approximately 70 to 400-500 nm (Moteki and Kondo, 2010; Kaspari et al., 2011), i.e., particles with diameter of >500 nm cannot be detected using an off-the-shelf SP2. Moteki and Kondo (2010) extended the upper limit of measurable rBC particle diameters to 850-900 nm (Moteki and Kondo, 2010; Ohata et al., 2011). More recently, an off-the-shelf instrument called the Single Particle Soot

Photometer Extended Range (SP2-XR; Droplet Measurement Technologies, USA), with measurable diameter range 50-800 nm, has become available. However, to our knowledge, no ice core rBC data produced by the SP2-XR have been published.

The extraction efficiency of the U5000AT ultrasonic nebulizer system at a flow rate of 0.19 mL min$^{-1}$ has been reported to be 10%–12% for the particle diameter range of approximately 200–500 nm; it decreases sharply for diameters >500 nm and decreases to approximately 2% for particles with diameter of 700 nm (Ohata et al., 2013; Mori et al., 2016). It also decreases for diameters < 200 nm (Ohata et al., 2013; Wendl et al., 2014; Mori et al., 2016). Thus, unless the size dependent extraction efficiency is carefully measured, as done by Moteki and Kondo (2010) and Ohata et al. (2011), measurements obtained using this nebulizer system could have large uncertainties not only in size distribution but also in mass concentration if the ice core samples contain BC particles with diameter of >500 nm, even if an SP2 with extended upper limit is used.

Modern snow and ice core samples from the Arctic, including Greenland, do contain substantial fractions of rBC particles with diameter of >500 nm (Mori et al., 2019). Similarly, modern snow from Antarctica also contains a consicerable proportion of rBC particles with diameters >500 nm (Kinase et al., 2020). If mass size distributions follow lognormal size distributions with mass median diameters <500 nm, mass concentrations for diameters > 500 nm can be estimated using lognormal fitting. However, non-lognormal mass size distributions with substantial concentrations of particles with diameters >850-900 nm have been reported for Arctic snow (Mori et al., 2019). Non-lognormal mass size distributions have also been observed in a Mt. Everest ice core, which contained substantial mass concentrations of rBC particles larger than the upper measurable diameter limit of 500 nm (Kaspari et al., 2011). Furthermore, bimodal mass size distributions with secondary modes diameters >500 nm have been reported for Antarctic snow (Kinase et al., 2020). Therefore, it is important to extend the measurable diameter range of rBC particles beyond 900 nm and to employ a nebulizer system with a high and size-independent extraction efficiency.

Mori et al. (2016) developed an improved technique for accurate measurement of the size distributions and concentrations of rBC particles with diameter between 70 nm and 4 μm in water samples. They used a Wide-Range SP2 (i.e., an SP2 modified to widen the measurable size range of rBC particles) and a Marin-5 high-efficiency concentric pneumatic nebulizer system (Teledyne CETAC, USA). For accurate, continuous, and high-resolution analyses of the concentrations and size distributions of rBC particles in polar ice cores, we combined the improved rBC measurement technique and a CFA system developed at the National Institute of Polar Research (NIPR). We used this system to analyse an ice core drilled at SIGMA-D in northwest Greenland (Matoba et al., 2015; Nagatsuka et al., 2021), following which we reconstructed the concentrations

and size distributions of rBC particles with diameter between 70 nm and 4 µm for the past 350 years. In this paper (called Part 1), we describe the coupled CFA-rBC measurement system and evaluate its performance. We compare the nebulizer efficiencies of Marin-5, APEX-Q, and U5000AT nebulizer systems; assess the stability of the efficiency of Marin-5 nebulizer system; examine the dispersion of the CFA-rBC signal; provide the evidence of minimal losses of rBC particles within the CFA-rBC system; and show examples of rBC size distributions. Since it is important to compare the data that our new rBC measurement system produced and the valuable data from the previous ice core rBC measurements, we also estimated the extent of underestimations in mass concentrations measured with the off-the-shelf SP2s. Using the new continuous high-resolution data, we investigated the seasonal variations in concentrations and size distributions of rBC particles originating from both anthropogenic and biomass burning emissions and their temporal changes. In a companion paper (Part 2), we discuss the derived results in detail.

## 2 Methods

### 2.1 Continuous flow analysis (CFA) system

To undertake high-resolution continuous analyses of ice cores, we developed a CFA system at NIPR. Figure 1 shows a schematic of the NIPR CFA system used to analyse the SIGMA-D core. It consists of a melting unit, debubbler unit, inductively coupled plasma–mass spectrometer (ICP-MS) unit, stable water isotope unit, microparticle unit, methane unit, and fraction collector unit in addition to an rBC unit. The rBC unit, ICP-MS unit, microparticle unit, and methane unit were added to an earlier version of the NIPR CFA system described by Dallmayr et al. (2016). The melting unit, debubbler unit, and the stable water isotope unit were the same as those used in the earlier version. Details of the melting unit, the ICP-MS unit, and the stable water isotope unit are provided in Appendix A. Although the NIPR CFA system includes a microparticle unit, methane unit, and fraction collector unit consisting of three fraction collectors, we do not discuss them further here because the data that they provided are not relevant. The specification and performance of each of these units will be reported elsewhere.

132

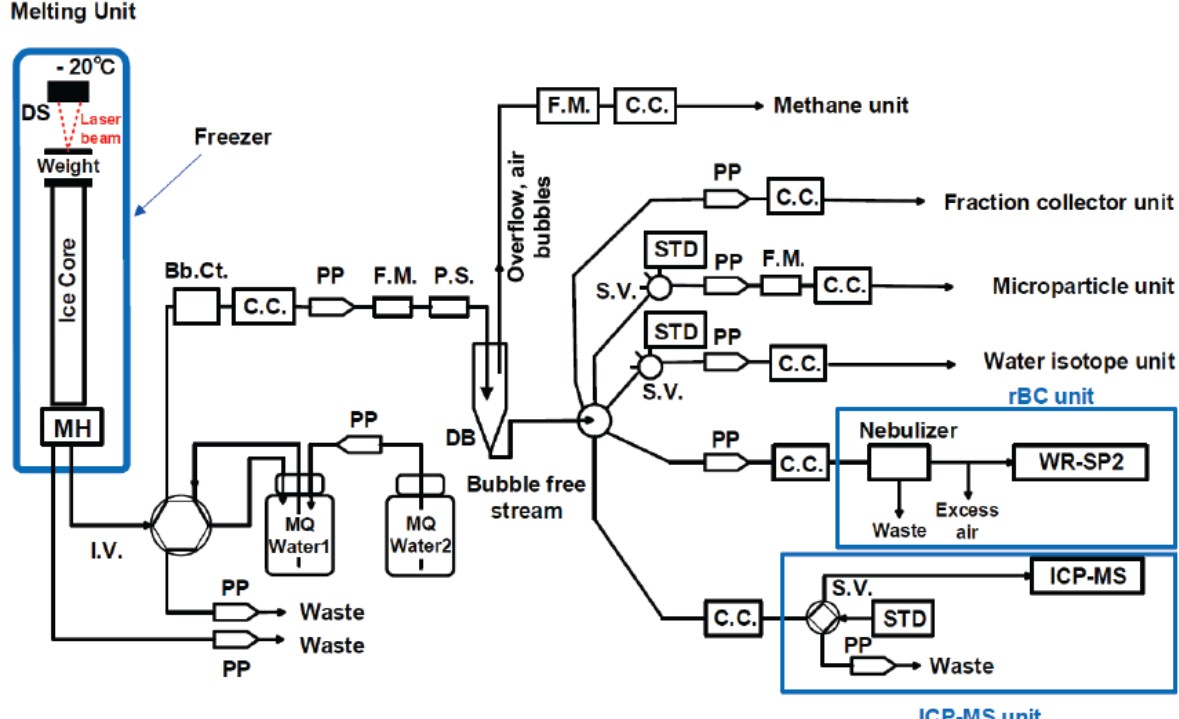

**Figure 1:** Schematic of the CFA system developed in this study.

DS: displacement sensor, MH: melt head, I.V.: injection valve, S.V.: selection valve, Bb.Ct.: bubble counter, C.C.: conductivity cell, PP: peristaltic pump, F.M.: flow meter, P.S. pressure sensor, DB: debubbler, STD: standard, MQ Water: ultra-pure water generated by a Milli-Q system.

**2.2 Refractory black carbon (rBC) unit**

We applied the improved technique developed by Mori et al. (2016) to the rBC unit of the NIPR CFA system. The rBC unit consists of a Wide-Range SP2 (Mori et al., 2016) and a concentric pneumatic nebulizer system (Marin-5, Teledyne CETAC, USA). The SP2 detects the incandescence signal from individual rBC particles induced by irradiation of an Nd-YAG laser (Stephens et al., 2003; Baumgardner, 2004; Schwarz et al., 2006). The off-the-shelf SP2 can detect rBC particles with diameter of between 70 and 400–500 nm, assuming rBC particle density of 1.8 g cm$^{-3}$ (Moteki and Kondo, 2010; Kaspari et al., 2011). The SP2 modified by Moteki and Kondo (2010) can meausure rBC particles with diameters between approximately 70 and 850-900 nm, whereas the off-the-shelf SP2-XR can measure rBC particles with diameters between approximately 50 and 800 nm. For the Wide-Range SP2, Mori et al. (2016) expanded the upper limit of the measurable diameter to 4 μm by modifying

the detection unit of the standard SP2. As a result, the Wide-Range SP2 can detect rBC particles with diameters of between approximately 70 nm and 4 µm. We used the "Standard SP2 Software" and the "Probe Analysis Package for Igor (PAPI)", both provided by DMT, to acquire and process the incandescent signal in binary data and convert it to text format. Then we used our original code to calculate the mass and size of BC particles.

The meltwater that passes through the debubbler unit is fed to the Marin-5 nebulizer system at a constant flow rate of 6.3 µL s$^{-1}$ by a peristaltic pump (REGLO Digital ISM596, ISMATEC, Germany) running at 7.50 rpm. We measured the flow rate before and after each CFA session. As the flow rate slightly (~5%) decreased after each CFA session, likely due to tube wear, we adjusted the flow rate of the peristaltic pump before the next CFA session. This approach allowed us to maintain a nearly constant flow rate with less than 5% variability. Under these conditions, no pulsed flow was observed. The Marin-5 nebulizer system was equipped with a MicroMist U-Series nebulizer AR30-1-UM05E (Glass Expansion, Australia). We used G3 Grade air as a carrier gas for the nebulizer. The flow rate of the carrier gas was 15.2 cm$^3$ s$^{-1}$ at standard temperature and pressure (i.e., 0 °C and 1013 hPa, respectively). The nebulizer system converts a fraction of the meltwater into water droplets that are immediately heated to 140 °C in a spray chamber, generating a mixture of rBC particles, non-rBC particles, and water vapor. After the non-aerosolized meltwater is removed via the drains, this mixture is cooled to 3 °C in a condenser, thereby removing the water vapor. Hence, only rBC and non-rBC particles are introduced to the Wide-Range SP2 at a flow rate of 12 cm$^3$ min$^{-1}$. The details of the Wide-Range SP2 and the Marin-5, together with assessment of their performance, have been reported by Mori et al. (2016).

To derive the relationship between the peak incandescence signal and the mass of each rBC particle (Stephens et al., 2003; Schwarz et al., 2006), we used fullerene soot (Alpha Aesar Inc., USA, Lot No. 20W054) as a standard material (Moteki and Kondo, 2010). We used an Aerosol Particle Mass Analyzer (Moteki and Kondo, 2010) Model 3601 (APM-II, KANOMAX, Japan) to extract fullerene soot particles with a mass range of 1.19–203 fg, corresponding to mass equivalent diameters of 100–600 nm. Following Mori et al. (2016), we produced two calibration curves for rBC masses below and above 10 fg, which corresponds to the mass equivalent diameter of 220 nm. Mass equivalent diameters of rBC particles were calculated assuming an rBC particle density of 1.8 g cm$^{-3}$ (Moteki and Kondo, 2010).

For accurate measurement of rBC particle size, the nebulizer efficiency and its size dependence must be known (Ohata et al., 2013; Mori et al., 2016). However, to the best of our knowledge, previous ice core studies using an SP2 rarely used nebulizer efficiency determined by measurements, except those conducted by Wendl et al. (2014), Lim et al. (2014) and Lim

et al. (2017). We determined nebulizer efficiency using Polystyrene Latex Sphere (PSL) suspensions with known number
concentrations (Ohata et al., 2011, 2013; Mori et al., 2016) for diameters of >200 nm. We used PSL particles supplied by two
manufacturers. The diameters of the PSL particles supplied by Polysciences Inc., USA (NIST Traceable Particle Size Standard),
were 207, 288, 505, 603, 707, 814, 1025, and 1537 nm, and the diameters of those supplied by Thermo Fisher Scientific Inc.,
USA, were 2000 and 3000 nm. For diameters of <200 nm, we used AquaBlack 162 (AB-162, Tokai Carbon Co. Ltd., Japan),
which is a laboratory standard for rBC particles suspended in water (Mori et al., 2016; Ohata et al., 2011; Ohata et al., 2013).
The number concentration of the PSL particles and that of the AquaBlack samples in the carrier gas were measured by the
Wide-Range SP2, and compared with those of the PSL suspensions and the B-162 suspensions, respectively, to calculate
nebulizer efficiency. Measurements of the PSL suspensions were performed with the SP2 laser currents lower than those for
rBC measurements. We repeatedly measured the efficiency of the Marin-5 nebulizer system over a ten year period.
Additionally, we measured the efficiency of the APEX-Q and U5000AT nebulizer systems. For the APEX-Q nebulizer system,
we used two types of nebulizers: the Conikal Nebulizer AR30-1-FC1ES (Glass Expansion, Australia) and the MicroMist U-
Series nebulizer AR30-1-UM05E (Glass Expansion, Australia), the latter being the same one used in the Marin-5 nebulizer
system.
Number and mass concentrations of rBC particles in the melted ice core samples were calculated using the nebulizer
efficiency (Mori et al., 2016). The combination of the Wide-Range SP2 and the Marin-5 nebulizer system provides a
measurable diameter range of 70 nm to 4 µm. With this rBC unit attached to the melting and debubbler units, we acquired
number concentrations, mass concentrations, and mass equivalent diameters of rBC particles every second. The detection
limits of rBC number and mass concentrations in water samples, determined as 3σ of the blank values, were approximately 10
counts·L$^{-1}$ and 0.01 µg·L$^{-1}$, respectively. The accuracy of the rBC number and mass concentrations in the water samples was
approximately 16%, which was derived from the measurement uncertainties of the peristaltic pump flow rate (±5%), nebulizer
flow rate (±5%), nebulizer efficiency (±10%), and rBC concentration in the carrier gas measured by the SP2 (±10%) (Mori et
al., 2016, 2021). The reproducibility of the number and mass concentrations for repeated measurements of the same melted
ice core and Arctic snow samples on two different days was usually better than 10% (Mori et al., 2019). For example, Mori et
al. (2019) demonstrated that the mass and number concentrations of rBC particles in a melted sample from the SIGMA-D core,
analysed on the day it was melted and again 21 months later, showed agreement within 5.6% and 4.4%, respectively. Mori et
al. (2019) further demonstrated that the changes in the mass and count median diameters were negligibly small in this sample.

Additionally, possible changes in the count median diameter during the nebulizing process were estimated to be only 2 nm for the fullerene soot, whose count median diameter was ~120 nm and whose mass concentration in water was 6.9–64 $\mu g \cdot L^{-1}$ (Mori et al., 2016). A similar value was estimated for the AB-162. These experimental results suggest that the shape of the rBC size distribution and the rBC mass concentration changed little during the nebulizer extraction process.

rBC particles could stick to the various components of the CFA system such as the melt head, debubbler, valves, conductivity cells, tubing, and nebulizer system, which could reduce the concentration and change the size distribution. We investigated whether losses of rBC particles occurred in the CFA system. We injected a melted surface snow sample from SIGMA-A (northwest Greenland) (Matoba et al., 2018) from above the centre hole of the melt head, and measured the concentration and size distribution of BC particles. We used the University of Copenhagen type melt head for this test. We also injected the same sample directly into the Marin-5 nebulizer system and measured the concentration and size distribution of rBC particles. We then compared the results of the two experiments to check whether any changes occurred that could be attributed to the CFA system.

## 2.3 Signal dispersion tests

The mixing of meltwater, which occurs in parts of the CFA system such as the melt head, debubbler, valves, conductivity cells, tubing, and nebulizer system, causes signal dispersion and reduces the resolution of the CFA data. To evaluate the signal dispersion, we examined the response of each unit by switching between injection of ultra-pure water and injection of standard solutions or melted ice core/snow samples at the melt heads (Bigler et al., 2011). The ultra-pure water, standard solutions, and melted ice core/snow samples were injected from above the centre hole of the melt heads. The ultra-pure water used in this study was made using a Milli-Q Advantage system (Merck Millipore, Germany). The samples used for the dispersion tests are listed in Table 1.

**Table 1. List of samples for signal dispersion tests**

| Measurement | Type of test samples |
|---|---|
| rBC | AquaBlack 162 (AB-162, Tokai Carbon Co. Ltd.) |
| ICP-MS | Surface snow from Dome Fuji, Antarctica, concentrated by heating |

| Stable water isotopes | A shallow ice core drilled at Dome Fuji, Antarctica (JARE52 core) |

**2.4 Processing and analyses of the SIGMA-D ice core**

A 222.7 m ice core was drilled at the SIGMA-D site (77.636° N, 59.120° W; 2100 m a.s.l.) in northwest Greenland in spring 2014 (Matoba et al., 2015). The annual mean air temperature and accumulation rate at the site were estimated to be −25.6 °C and 0.23 w eq yr$^{-1}$ (Nagatsuka et al., 2021), respectively. In the field, the top 175.77 m of the core was divided into two vertical sections (Sections A and B).

Section A was kept frozen and transported to NIPR in Japan. We analysed the depth interval between 6.17 and 112.87 m of this section using the CFA system described in Sect. 2.1 and 2.2. The top 6.17 m of this section was too fragile to be analysed with the CFA system; hence, we manually cut segments of approximately 0.1 m. These discrete samples were decontaminated in a cold room (−20 °C) using a precleaned ceramic knife, and then placed in powder-free plastic bags. They were then melted and transferred to precleaned glass and polypropylene bottles in a class 10,000 clean room. The samples in glass bottles were analysed for stable water isotopes and rBC, whereas those in polypropylene bottles were analysed for six elements using an ICP-MS. Analyses of stable water isotopes and six elements are described in Appendix B1. The rBC was analysed using a Wide-Range SP2 (Mori et al., 2016) and a concentric pneumatic nebulizer system (Marin-5, Teledyne CETAC, USA), i.e., the same as those in the NIPR CFA system. The setting and analytical conditions of the Wide-Range SP2 and Marin-5 were similar to those described in Sect. 2.2. Concentrations and diameters of rBC particles were calibrated in the same way as described in Sect. 2.2.

Section B was cut in the field into 0.06–0.12 m long vertical segments for the top 5 m of the core, 0.05–0.08 m long segments for depths of 5–12 m, and approximately 0.05 m long segments for the depth interval between 12 and 112.87 m. Each segment was placed in a plastic bag, melted, and transferred to a precleaned polypropylene bottle in the field. The discrete samples contained in the polypropylene bottles were refrozen in the field, transported to Japan, and kept frozen until analysis, whereupon they were melted and analysed for major ions and stable water isotopes (Nagatsuka et al., 2021). Analyses of the discrete samples from Section B are described in Appendix B2.

**3 Results and Discussion**

**3.1 Nebulizer efficiency**

Figure 2 shows the efficiency of the Marin-5 nebulizer system for different flow rates of meltwater. As previously reported
(Mori et al., 2016), nebulizer efficiency depends on flow rate. For three flow rates—0.19, 0.38, and 0.48 mL·min$^{-1}$—the
efficiency was almost constant for diameters of <2000 nm, and it declined linearly with diameter for diameters >2000 nm, as
reported by Mori et al. (2016). For a flow rate of 0.38 mL·min$^{-1}$, which is the flow rate used in the NIPR CFA system, the
efficiency was 34.2% ± 8.0% for particles with diameter of <2000 nm, and it declined linearly with diameter for diameters of
2–4 µm. The efficiency of Marin-5 was slightly higher than that of APEX-Q for the PSL with diameters between 200 and 700
nm at a flow rate of 0.38 mL min$^{-1}$ (Fig.C1(a)). Repeated measurements of the Marin-5 efficiency over a ten-year period (Fig.
3) indicate that the nebulizer efficiencies remained stable over time, despite some fluctuation around the regression lines. For
particles with diameters < 2 µm, the variability was ±8 %, which does not significantly affect the rBC data. Consequently, we
applied the same nebulizer efficiency values across all CFA sessions. Additionally, we validated the stability of our WR-
SP2/nebulizer system by repeatedly measuring the rBC mass and number concentrations in the same samples, as demonstrated
by Mori et al. (2019). In contrast to the Marin-5 nebulizer system, the U5000AT nebulizer system exhibited size- and time-
dependent efficiency (Fig. C2(b)).

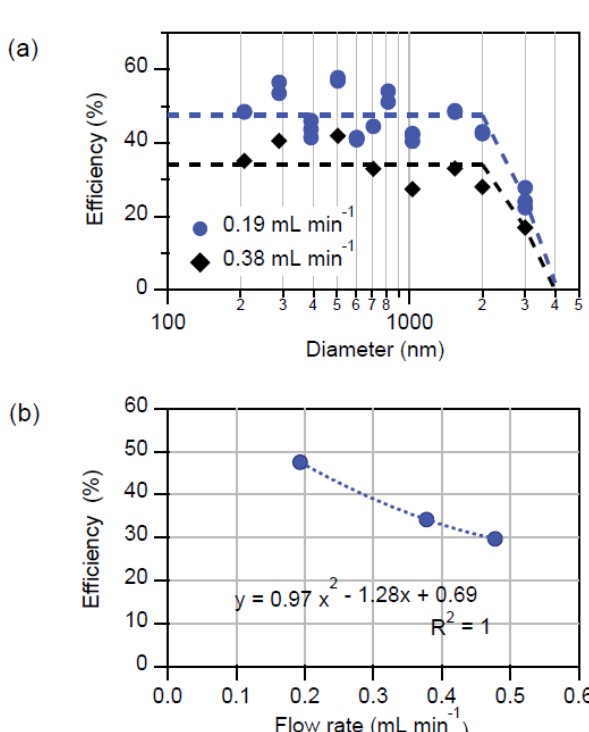

**Figure 2:** Dependence of Marin-5 nebulizer efficiency on (a) BC diameter for two flow rates and (b) flow rate for

BC diameter of <2 µm.

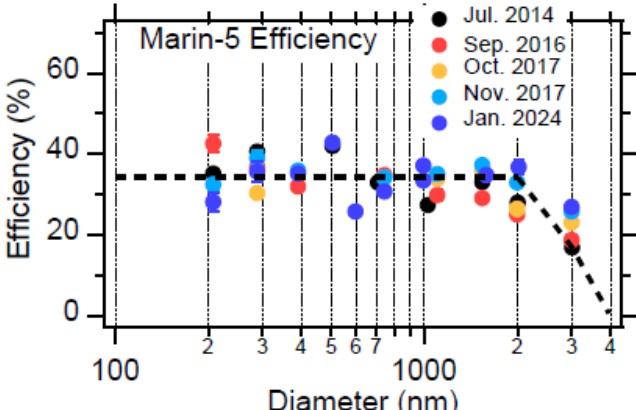

**Figure 3:** Repeated measurements of Marin-5 nebulizer efficiency over ten years for a flow rate 0.38 mL L$^{-1}$.


**3.2 Signal dispersion**
Figure 4 displays the results of dispersion tests for δ$^{18}$O, Na, and rBC. We defined two types of response times: (1) the time
(t1) required for transition from 10% of the standard (or ice core/snow sample) value to 90% of the standard (or ice core/snow

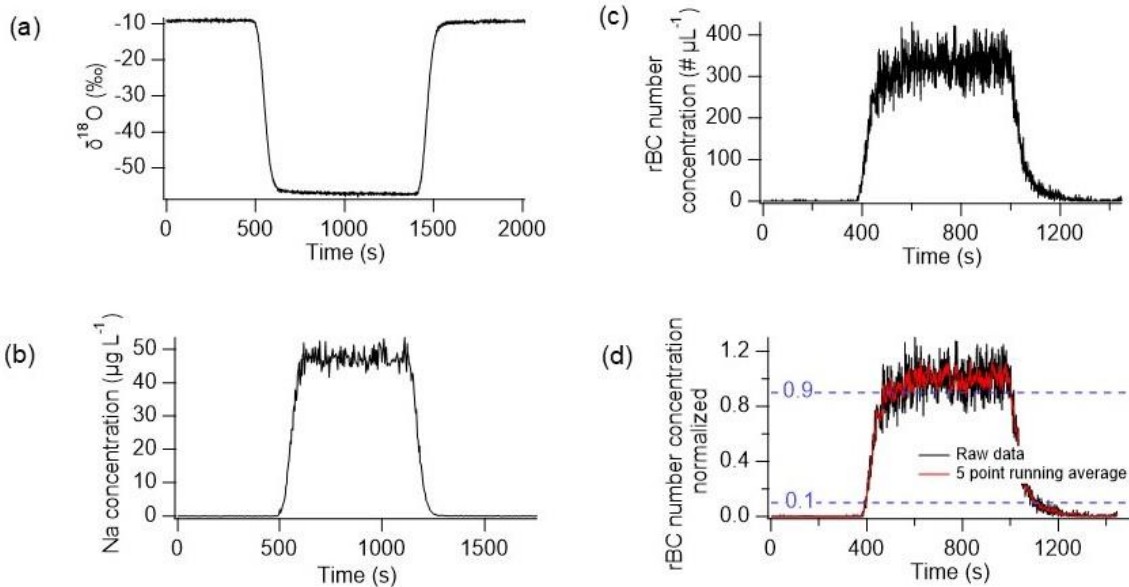

**Figure 4:** Results of dispersion tests: (a) δ$^{18}$O, (b) Na concentration, (c) rBC number concentration, and (d) normalized rBC number concentration. Black and red lines represent raw data and 5 point running averages, respectively. Blue dotted lines show 0.1 and 0.9 levels.

sample) value, and (2) the time (t2) required for transition from 90% of the standard (or ice core/snow sample) value to 10% of the standard (or ice core/snow sample) value. The baseline was determined as the value for Milli-Q water. Response times t1 and t2 depend on how the data are smoothed owing to noise in the data signal. Table 2 shows examples of t1 and t2 when the data are smoothed by taking 5-point running means. Neither t1 nor t2 depends on the standard (or ice core/snow sample) concentrations or values (Bigler et al., 2011). For rBC, we present normalized values together with concentrations in Fig. 4 to illustrate how we determined t1 and t2. We converted t1 and t2 to depth intervals L1 and L2, respectively, assuming a constant melt speed of 30 mm s$^{-1}$. In Table 2, we list the averages of L1 and L2 for a rise of 10%–90% and decay of 90%–10%, respectively. L1 and/or L2 are often defined as the depth resolution of a CFA system (Bigler et al., 2011; Erhardt et al., 2023; Grieman et al., 2022). This definition gives a depth resolution of 35-40 mm for the $\delta^{18}$O, Na, and rBC data over the depth interval between 6.17 and 112.87 m. However, the resolution of our CFA system is better than these values suggest. We could resolve two peaks located at distances closer than the resolution defined in this way. For $\delta^{18}$O, Na, and rBC, peaks 10 mm apart are usually resolved, although peak heights may be slightly reduced for peaks that are less than 35-40 mm apart. For rBC and Na, L2 is slightly greater than L1, indicating that the melting direction affects the CFA signal (Breton et al., 2012). The CFA signal for rBC and Na might not be symmetrical, even if a concentration peak is symmetrical along the core depth (Breton et al., 2012). Conversely, $\delta^{18}$O shows similar L1 and L2 values, indicating that melting direction does not affect the CFA signal.

In addition to the mixing that occurs in the debubbler, valves, conductivity cells, tubing, and nebulizer systems, there is also mixing between the meltwater from the center of the ice sample and the meltwater from the ice on outside the inner wall of the melt head. However, due to the very short distance and very small dead volume within the melt head (using a 26 x 26 mm square-shaped melt head as described by Bigler et al. (2011)), this mixing is negligible compared to the mixing that occurs in other parts of the CFA system. Therefore, the signal dispersion observed in this study provides a reliable representation of the dispersion caused by the entire CFA system. Additionally, the stratigraphy of the SIGMA-D core was nearly horizontal, resulting in minimal mixing of ice from different ages.

**Table 2 Results of dispersion tests**

| | University of Copenhagen type melt head | University of Maine type melt head |
|---|---|---|

| | t1 (s) | t2 (s) | L1 (mm) | L2 (mm) | Average of L1 & L2 (mm) | t1 (s) | t2 (s) | L1 (mm) | L2 (mm) | Average of L1 & L2 (mm) |
|---|---|---|---|---|---|---|---|---|---|---|
| $\delta^{18}O$ | 78 | 75 | 39 | 37.5 | 38.3 | 75 | 81 | 37.5 | 40.5 | 39 |
| rBC number concentration | 67 | 90 | 33.5 | 45 | 39.3 | 105 | 124 | 52.5 | 62 | 57.3 |
| Na concentration | 66 | 74 | 33 | 37 | 35 | 57 | 89 | 28.5 | 44.5 | 36.5 |


### 3.3 Minimal losses of BC particles in the NIPR CFA system

Figure 5 and Table 3 present the results of rBC loss tests. The sample injected at the melt head, which then flowed through the
CFA system, produced mass and number size distributions of rBC particles consistent with those derived following direct
injection. The mass and number concentrations of rBC particles injected at the melt head were 94% and 102% of those
determined following direct injection. Thus, the rBC concentrations of the two types of injections agreed within the bounds of

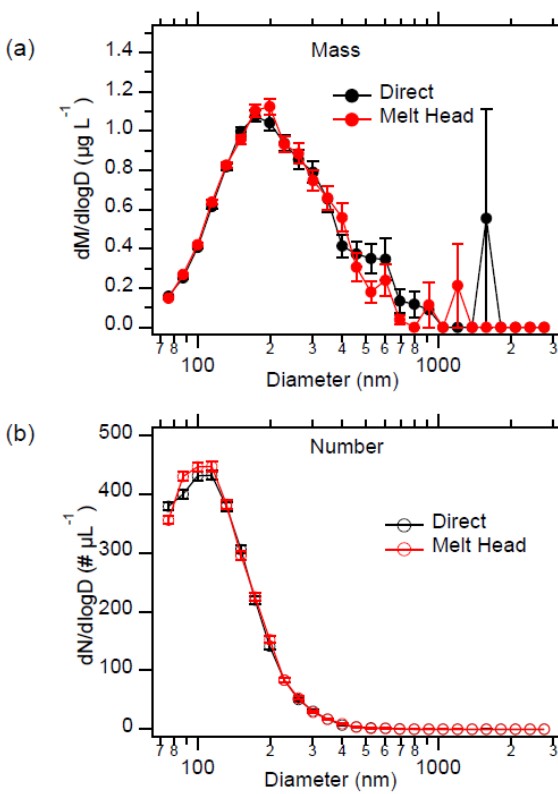

**Figure 5:** Comparison of direct injection of a surface snow sample collected at SIGMA-A to Marin-5 and injection at the melt head. (a) Mass and (b) number size distributions of rBC shown for direct and melt head injections. Error bars indicate ±1σ of a Poisson distribution.

uncertainty of the BC measurements. Therefore, we can conclude that minimal loss of rBC particles occurs in the NIPR CFA system. The good agreement between injection at the melt head and the direct injection also supports the reliability and reproducibility of the NIPR CFA-rBC system.

**Table 3 Results of rBC loss test using a surface snow sample from SIGMA-A, northwest Greenland**

| | Melt head blank | Injection at melt head | Direct injection | Ratio of injection at melt head/direct injection |
|---|---|---|---|---|
| rBC mass concentration ($\mu g\ L^{-1}$) | 0.004 | 0.623 | 0.660 | 0.944 |
| rBC number concentration ($\#\ L^{-1}$) | 0.1 | 175.8 | 173.0 | 1.016 |

**3.4 High-resolution rBC data from the SIGMA-D ice core**

Figure 6 displays the raw data of BC mass and number concentrations acquired using the CFA system at 1 s interval
corresponding to a depth interval of 0.0005 m, together with the 10 mm averages of the data. The ice-core chronology
determined by Nagatsuka et al., (2019) with a slight modification (Goto-Azuma et al., 2024) is shown in Fig. 6. The raw mass
concentration data frequently exceeded 50 $\mu g \cdot L^{-1}$. However, as can be deduced from the differences in mass concentrations
and number concentrations (Fig.6 (a) and (b)) and their enlarged extracts (Fig.6 (c) and (d), respectively), the sporadic high
concentration peaks in the raw mass concentration data could have been formed by only a small number of large BC particles,
which would result in the noise in the data. To reduce the noise, we calculated the 10 mm averages of the data, corresponding
to a 1–2 week interval depending on the depth of the core. Averaging the mass concentrations over 10 mm intervals effectively
filtered out data noise, while still preserving the large peaks, albeit with slightly reduced amplitudes (Fig. 6(a) and (c)). The
10 mm averages of the mass concentrations often exceeded 10 $\mu g \cdot L^{-1}$. The prominent peaks in mass and number concentrations

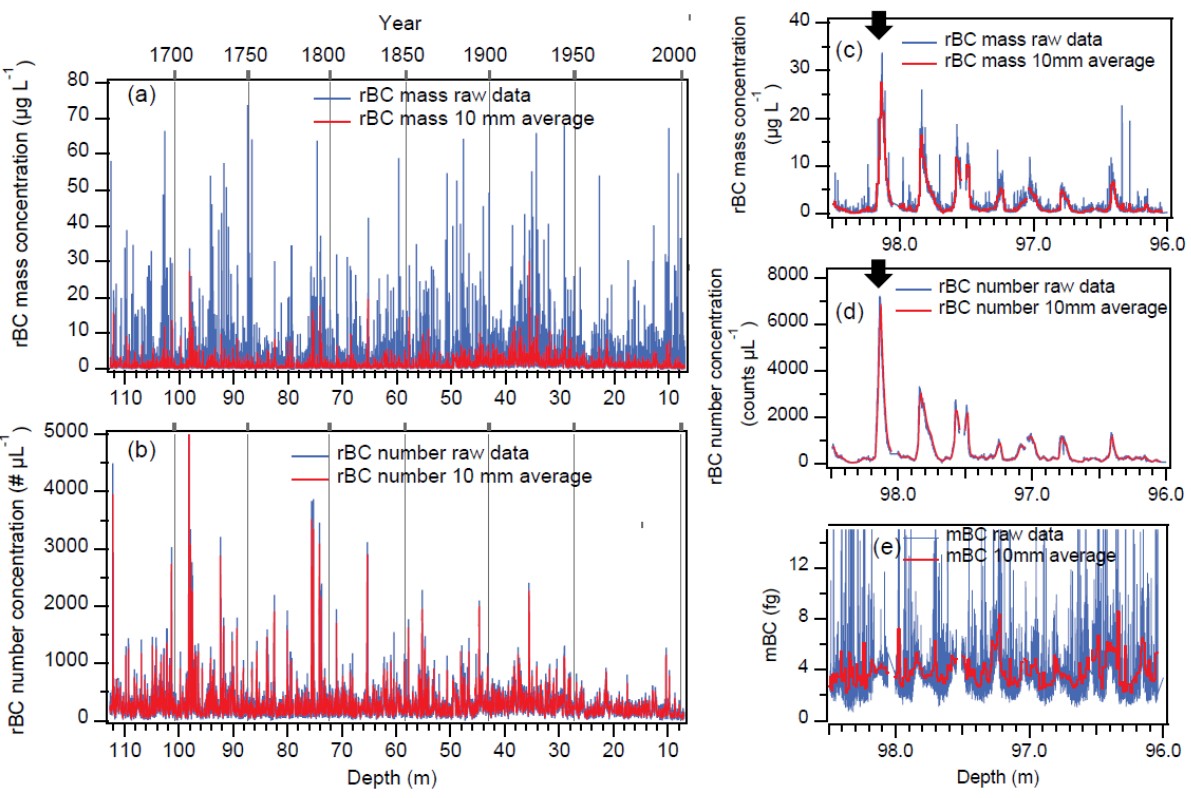

**Figure 6:** (a) Mass and (b) number concentrations of rBC in the SIGM-D core. (c) and (d) are enlarged extracts

of (a) and (b), respectively. (e) mBC (average mass of rBC particles) for the same depth interval as (c) and (d).

Raw data and 10 mm averages of the raw data are shown in blue and red, respectively. The arrows (c) and (d)

denote the summer of 1710.

around 98.1 m correspond to the summer of 1710, when rBC particles from a significant biomass burning event were deposited
at the SIGMA-D site (Goto-Azuma et al., 2024).
The upper limit of the measurable rBC diameters would affect the rBC mass concentrations if the ice core samples
contain a large proportion of large particles. As described in Sect. 2.2, the upper limit of the NIPR rBC unit is 4 μm, whereas
the upper limit of a measurement system using the off-the-shelf SP2 is 400-500 nm and that of a measurement system using
an extended range SP2 and the off-the-shelf SP2-XR is 800-850 nm. If a measurement system uses a nebulizer system such as
the U5000AT ultrasonic nebulizer system (Teledyne CETAC, USA), which was used in many previous studies, nebulizer
efficiency is drastically reduced for diameters greater than approximately 500 nm (Mori et al., 2016), which would lead to
underestimation of rBC mass concentrations if the ice core contains a large proportion of rBC particles with diameter of >500
nm even if an extended range SP2 is used. We calculated the number and mass size distributions of rBC particles averaged

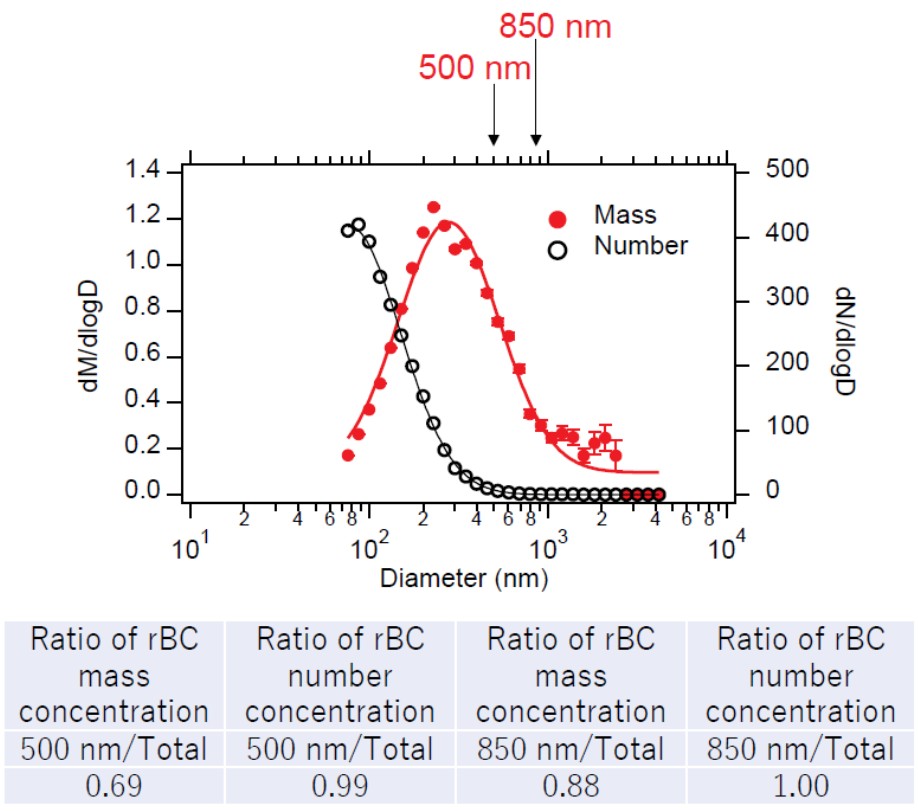

| Ratio of rBC mass concentration 500 nm/Total | Ratio of rBC number concentration 500 nm/Total | Ratio of rBC mass concentration 850 nm/Total | Ratio of rBC number concentration 850 nm/Total |
|---|---|---|---|
| 0.69 | 0.99 | 0.88 | 1.00 |

**Figure 7:** Averaged number (black) and mass (red) size distribution of rBC particles for the period 2003–2013, respectively. Error bars indicate $\pm 1\sigma$ of a Poisson distribution. The table shows ratios of concentrations for upper limits of 500 and 850 nm to total concentrations.

over different periods. As an example, the 11-year mean number and mass size distributions for 2003–2013, derived from
analyses of the discrete samples, are plotted in Fig. 7. It is evident from Fig. 7 that the total number concentrations of rBC
particles would have been affected little by the upper limits of the measurable BC diameters, which were approximately 400-
500 nm in previous studies and 850-900 nm if an extended range SP2 was used. In contrast, the mass concentrations would
have been underestimated by 31% and 12% for upper limits of 500 and 850 nm, respectively.

Figure 8 displays additional examples of mass size distributions of rBC particles for months with significant rBC

concentration peaks. Given that the upper limit of measurable rBC diameter is 500 nm, mass concentrations during the
summers of 1710 and 1863, and the winters of 1916/17 and 1935/36 would have been underestimated by 8, 43, 26, and 36 %,
respectively. The mass size distribution, and consequently the degree of underestimation, varied over time. We calculated the
average mass of rBC particles (mBC), by dividing the mass concentration by the number concentration, which serves as one
of the rBC size parameters. Fig. 6(e) illustrates an example of the mBC variability with depth, indicating seasonal changes in
rBC size distribution. A companion paper (Part 2, Goto-Azuma et al., 2024) further investigated the temporal variability in
rBC size distribution. As rBC size distribution changes over time, the underestimation ratio cannot be assumed to be constant.
Therefore, it is crucial to extend the measurable rBC diameters beyond 500 nm, desirably beyond 800-850 nm.

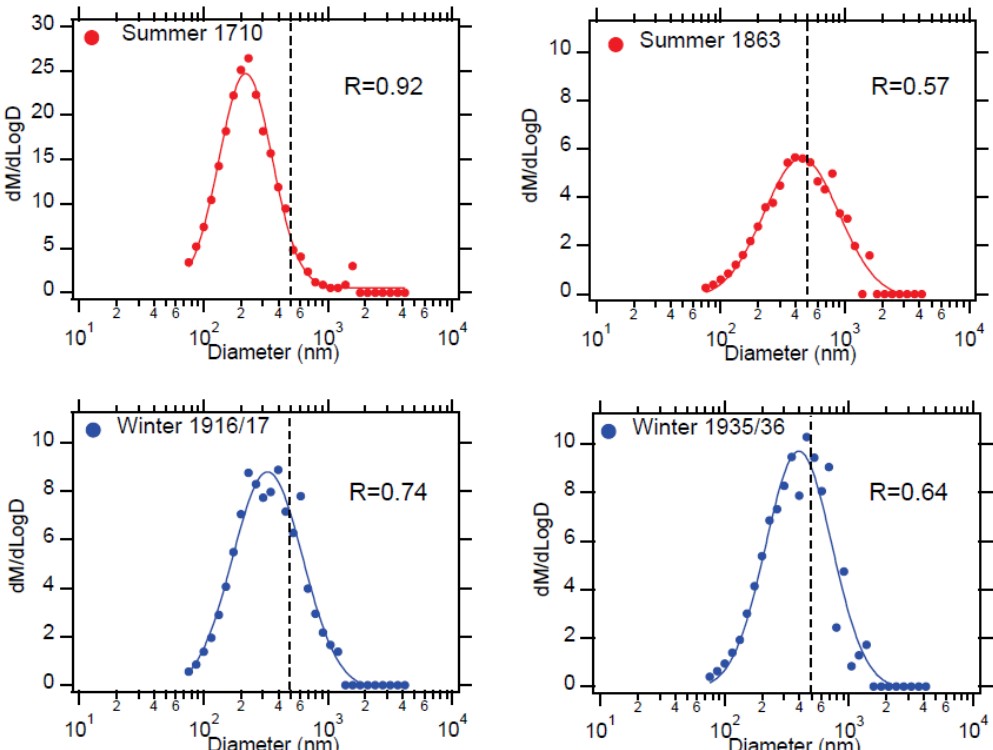

**Figure 8:** Examples of mass size distributions of rBC particles for summer and winter months showing high rBC concentrations. (a) Summer months of 1710, shown by the arrows in Fig. 6 (a) and (b). (b) Summer months of 1863. (c) Winter months of 1916/17. (d) Winter months of 1935/36. Summer and winter months correspond to approximately May-July and December-February, respectively (Goto-Azuma et al., 2024). The dotted lines show the upper limit of measurable rBC diameter (500 nm) for off-the-shelf SP2. R denotes the ratio of rBC mass concentration for diameter <500 nm to the total rBC mass concentration.

To examine the impact of large rBC particles in the SIGMA-D ice core, the rBC mass concentrations averaged for

10 mm intervals, assuming different upper limits, were calculated from the size distribution data, and plotted in Fig. 9a. In Fig.
9b, the ratios of the rBC mass concentrations for different upper limits versus the total rBC mass concentrations are shown.
Figure 9b shows that the off-the-shelf SP2 combined with a size-independent high-efficiency nebulizer system such as the
Marin-5 or the APEX-Q nebulizer systems, which would give an upper limit of 500 nm, would occasionally underestimate the
rBC mass concentration by 30-40% or more. Even an extended range SP2, when combined with a size-independent high-
efficiency nebulizer system, could occasionally underestimate the rBC mass concentration by 20% or more. If we use a
nebulizer system such as the U5000AT, underestimation would be even greater, though difficult to quantify due to its size-

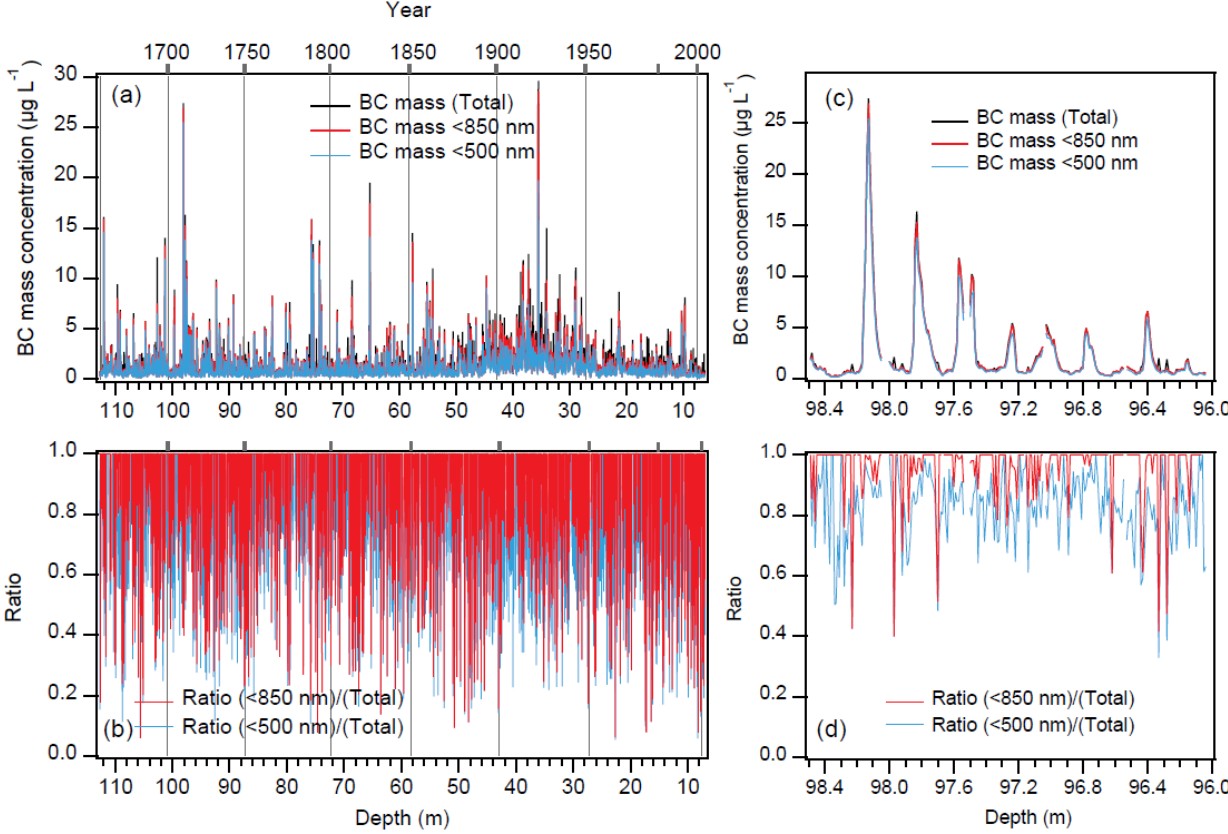

**Figure 9:** Comparison of rBC mass concentration (10 mm averages) in the SIGMA-D core for different upper limits of measurable rBC diameters. (a) Total concentration measured in this study (upper limit: 4 um), concentration for upper limit of 850 nm, and concentration for upper limit of 500 nm are displayed in black, red, and blue colors, respectively. (b) Ratio of rBC mass concentration for upper limit of 850 nm (red) and 500 nm (blue) to total concentration. (c) and (d) are enlarged extracts of (a) and (b), respectively.

and time- dependent efficiency. Figure 10 presents histograms of the ratios of rBC mass concentrations for upper limits of 500
and 850 nm. For the upper limit of 500 nm, 67% of the 10 mm averages account for <90% of the total rBC mass concentrations;
whereas for the upper limit of 850 nm, 15% of the 10 mm averages account for <90% of the total rBC mass concentrations.




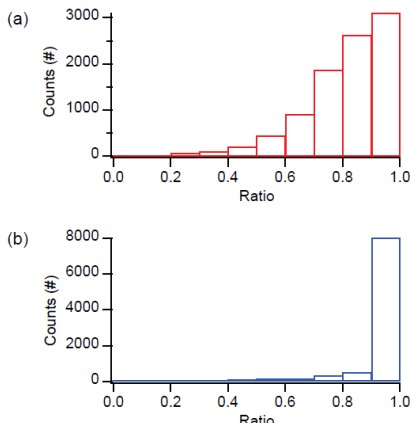

**Figure 10:** Histograms of underestimation for 10 mm averaged data. Horizontal axis represents the ratio of mass concentration for an upper limit of (a) 500 nm and (b) 850 nm. Vertical axis represents the number of 10 mm averaged data in each ratio bin.


**4. Conclusions**
We developed a CFA system and incorporated an rBC unit that uses the improved rBC measurement technique developed by
Mori et al. (2016). The CFA system can acquire continuous and high-resolution measurements of the number and mass
concentrations of rBC, and the size distribution of rBC particles, in addition to stable water isotopes ($\delta^{18}$O and $\delta$D), six elements
($^{23}$Na, $^{24}$Mg, $^{27}$Al, $^{39}$K, $^{43}$Ca, and $^{56}$Fe), microparticles, electrical conductivity, and methane. There were minimal losses of rBC
particles within the NIPR CFA system. We analysed the SIGMA-D ice core retrieved from northwest Greenland using this
newly developed system. If we define the depth resolution as the average of the rise of 10%–90% and decay of 90%–10% of
the CFA signal, the resolutions were 38, 39, and 35 mm for $\delta^{18}$O, rBC, and Na, respectively. These depth resolutions correspond
to the temporal resolutions of 0.08–0.16, 0.11–0.23, and 0.07–0.15 years for $\delta^{18}$O, rBC, and Na, respectively, depending on
depth. However, we could usually resolve two peaks that were approximately 10 mm apart, corresponding to 1-2 weeks
depending on depth. We were able to analyse monthly resolved rBC data as described in the companion paper, i.e., Part 2 of
our study on rBC in the SIGMA-D core (Goto-Azuma et al., 2024).
The Wide-Range SP2 and the Marin-5 nebulizer system allowed analysis of rBC particles with diameter between
approximately 70 nm and 4 μm, contrasting with the analysis of rBC particles with diameter of between 70 and 400-500 nm
reported in previous ice-core studies. This enabled us to reconstruct accurate mass concentrations and size distributions of rBC
particles, together with their temporal changes (Goto-Azuma et al., 2024), which could contribute to estimation of the impacts

of rBC on the radiation budget and cloud microphysics. Using the size distribution data, we estimated the extent of underestimation that would result from using (1) an off-the-shelf traditional SP2, which can measure rBC particles with diameters <500nm,, and (2) an SP2 modified by Moteki and Kondo (2011) or an off-the-shelf SP2-XR which can measure rBC particles with diameter <850 nm. In both cases, we assumed size-independent efficiency for nebulizer systems such as the Marin-5 and the APEX-Q. For (1), approximately 67% and 37 % of the 10 mm averaged data from the SIGMA-D core accounted for <90% and <80% of the total rBC mass concentrations, respectively. For (2), approximately 15% and 10% of the 10 mm averaged data from the SIGMA-D core accounted for <90% and <80% of the total rBC mass concentrations, respectively. The extent of the underestimation depends on depth and thus on the age of the core. For the period 2003–2013, (1) and (2) would lead to underestimation of the averaged mass concentration by 31% and 12%, respectively. For large concentration peaks resulting from significant boreal forest fires and anthropogenic inputs, underestimation would frequently exceed 40%.

Although few ice core studies have considered the size distribution of rBC and estimated the extent of underestimation of rBC mass concentrations, the present-day snow from Svalbard (Mori et al., 2019) and an ice core from Mt. Elbrus in the western Caucasus Mountains (Lim et al., 2017) do contain non-negligible amounts of rBC particles with diameter of >500 nm or 850 nm. Since the size distributions do not always follow the lognormal distributions, the improved method for accurate measurement of rBC mass concentrations should be employed to properly constrain aerosol models.

**Appendix A: Details of the NIPR CFA system**

An ice core sample (cross section: 34 mm × 34 mm, length: ~0.5 m) was placed on a melt head inside a freezer. An 850 g weight was placed on top of the ice sample to allow stable melting. Before the ice core sample was completely melted, another similarly sized ice core sample was stacked on top of the first sample to maintain continuous melting of the ice samples. To promote melting, heaters are inserted into the melt head (Bigler et al., 2011; Osterberg et al., 2006). In the earlier NIPR CFA system, we used a melt head developed at the University of Maine (Osterberg et al., 2006). However, in this study, we used a melt head similar to the one developed at the University of Copenhagen by Bigler et al. (2011) for the depth interval between 11.3 and 112.8 m of the SIGMA-D core. The University of Maine type melt head, designed principally for use in firn core analyses, is not airtight. For methane analysis, we had to use an airtight melt head such as the one used by Bigler et al. (2011). For the depth interval between 6.17 and 11.3 m of the SIGMA-D core, we used the University of Maine type melt head

(Dallmayr et al., 2016; Osterberg et al., 2006) to reduce water percolation through the porous firn caused by capillary action
(Osterberg et al., 2006). For depths < 49.3 m, methane measurement was not performed.

The depth of an ice core sample was assigned using a laser positioning sensor (LKG-G505, Keyence, Japan), which

determined the distance from the sensor to the top of the weight (Dallmayr et al., 2016). A typical melt speed, regulated by the
voltage applied to the heaters in the melt head, was 30 mm min$^{-1}$. The depth resolution of the laser positioning sensor with this
melt speed was approximately $0.3 \pm 0.1$ mm. The meltwater collected in the contamination-free inner part of the melt head is
drawn through perfluoroalkoxy alkane tubing, an injection valve, and the debubbler unit by a peristaltic pump (Minipuls3 MP-
2, Gilson, USA). Following removal of air bubbles by the debubbler unit, the meltwater is introduced to the different
measurement units and to the fraction collector unit using peristaltic pumps (Reglo Digital ISM596, ISMATEC, Germany).
Before each unit, an electrical conductivity cell (conductivity meter Model 1056, Amber Science Inc., USA) is placed as close
as possible to the unit to synchronise the depths of the ice core data acquired by the different measurement units and the depths
of the meltwater samples collected by the fraction collector unit (McConnell et al., 2002; Dallmayr et al., 2016). A length of
approximately 7 m of the ice core was melted once or twice a week. The lengths of the tubing between (1) the melt head and
the debubbler, (2) the debubbler and the ICP-MS unit, (3) the debubbler and the water istotope unit, and (4) the debubbler and
the rBC unit were approximately 1 m, 3 m, 1.2 m, and 1.5m, respectively. The inner diameters of the tubing for the ICP-MS
unit, water isotope unit, and rBC unit were 0.03, 0.02, and 0.03 inches, respectively.

The ICP-MS unit consists of an ICP-MS (7700 ICP-MS, Agilent Technologies, USA) including a nebulizer system.

The elements $^{23}$Na, $^{24}$Mg, $^{27}$Al, $^{39}$K, $^{43}$Ca, and $^{56}$Fe were each measured at a 3.00 s interval. Additionally, $^{89}$Y was measured
at a 3.00 s interval to check the stability of the ICP-MS. Data acquisition times for $^{23}$Na, $^{24}$Mg, $^{27}$Al, $^{39}$K, $^{43}$Ca, $^{56}$Fe, and $^{89}$Y
were 0.02, 0,1, 0.2, 0.1, 2.27, 0.252, and 0.044 s, respectively. We used mainly $^{23}$Na data to date the core. The concentration
of each of the elements was calibrated both before and after the CFA measurements of the day using a multi-element standard
solution (XSTC-331, Spex CertiPrep, USA) diluted with ultra-pure water (Milli-Q water, Milli-Q Advantage, Merck
Millipore, Germany). The detection limit, defined as [3σ of the blank value + the intercept of the calibration line], of $^{23}$Na
is 0.5 µg L$^{-1}$.

The stable water isotope unit is essentially same as that used by Dallmayr et al. (2016). It consists of a vaporization

module (Gkinis et al., 2011; Dallmayr et al., 2016), and a wavelength-scanned cavity ring-down spectrometer (L2130-i or
L2120-i, Picarro Inc., USA). The Picarro L2130-i was used for the depth interval between 107.3 and 49.3 m, while the
Picarro L2120-I was used for the remaining depths. We calibrated the spectrometer by analysing three sets of laboratory
water isotope standards after the CFA measurements of the day. These laboratory standards were calibrated with VSMOW2
and SLAP2 standards purchased from the International Atomic Energy Agency. Details of calibrations and the performance
of the stable water isotope unit have been described in a previous study (Dallmayr et al., 2016). Both the L2130-i and L2120-
i demonstrated sufficient stability during the 4-5 hours of a daily CFA session, confirmed by Mill-Q water runs before and
after the CFA measurements. The good agreement between the CFA data (from Section A of the SIGMA-D core) and
discrete sample data (from Section B of the core) also confirms the stability of both Picarros (Goto-Azuma et al., 2024).

**Appendix B: Analyses of discrete samples**
**B1 Discrete samples from Section A of SIGMA-D core**
From the top 6.17 m of Section A of the SIGMA-D core, discrete samples were prepared (Sect. 2.4). The samples in glasss
bottles were analysed for stable isotopes of water using a near-infrared cavity ring-down spectrometer (L2120-i, Picarro, Inc.
USA), a high-precision vaporizer (A0211, Picarro Inc., USA), and an autosampler (PAL HTC9 - xt - LEAP, LEAP
Technologies, USA). The precision of determination was ±0.05‰ for $\delta^{18}O$. The samples in polypropylene bottles were
analysed for six elements ($^{23}$Na, $^{24}$Mg, $^{27}$Al, $^{39}$K, $^{40}$Ca, and $^{56}$Fe) with an ICP-MS (7700 ICP-MS, Agilent Technologies, USA)
in a class 10,000 clean room at NIPR.

**B2 Discrete samples from Section B of SIGMA-D core**
Samples from depths above 61.2 m were analysed for $Na^+$, $K^+$, $Mg^{2+}$ and $Ca^{2+}$, $Cl^-$, $NO_3^-$, and $SO_4^{2-}$ with two ion
chromatographs (ICS-2100, Thermo Fisher Scientific, USA) at Hokkaido University, Japan, whereas samples from depths
between 61.2 and 112.87 m were analysed for $NH_4^+$, $Na^+$, $K^+$, $Mg^{2+}$, $Ca^{2+}$, $Cl^-$, $NO_3^-$, and $SO_4^{2-}$ with two ion chromatographs
(ICS-2000, Thermo Fisher Scientific, USA) at NIPR. The limit of detection of $Na^+$ measured at Hokkaido University was 10
$\mu g \cdot L^{-1}$, and that measured at NIPR was 0.2 $\mu g \cdot L^{-1}$. Stable water isotopes were analysed for all samples from Section B using
a near-infrared cavity ring-down spectrometer (L2130-i, Picarro Inc., USA) and a high-throughput vaporizer (A0212, Picarro
Inc., USA) at Hokkaido University. The precision of determination was ±0. 1 ‰ for $\delta^{18}O$. The good agreement between the
CFA data and discrete sample data from Sections A and B, respectively, ensured the high quality of the CFA data. For dating
purposes, tritium concentrations were measured using a liquid scintillation counter (LSC-LB3; Aloka Co. Ltd., Japan) at 0.05
m intervals for the depth interval of 19.15–26.47 m (Nagatsuka et al., 2021).

**Appendix C**

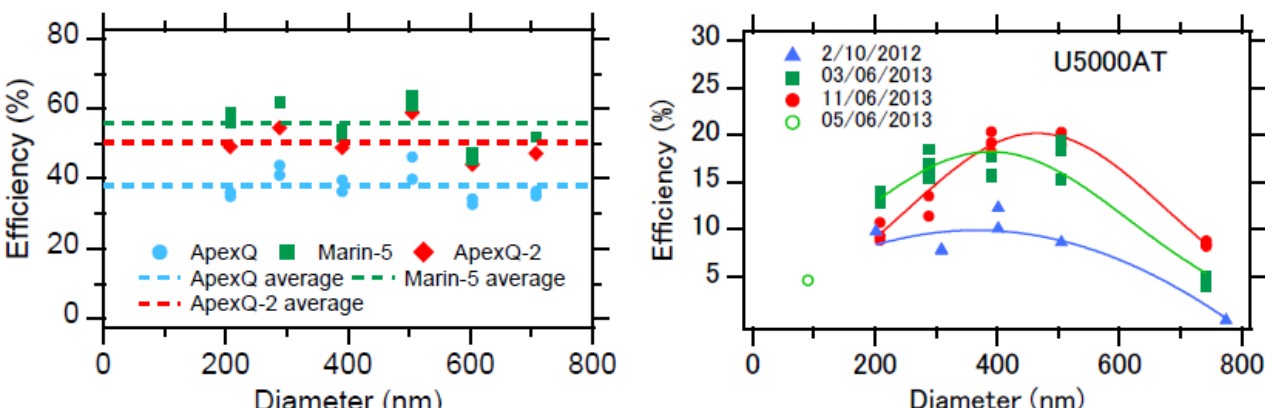

**Figure C1:** (a) Comparison of Marin-5 and APEX-Q nebulizer efficiency for a flow rate of 0.19 mL min[-1]. A MicroMist

U-series AR30-1-UM05E (Glass Expansion, Australia) was used for the Marin-5 nebulizer system. On the other hand, two

types of nebulizers, a Conikal Nebulizer AR30-1-FC1ES (Glass Expansion, Australia) and a MicroMist U-Series nebulizer

AR30-1-UM05E (Glass Expansion, Australia) were used for the Apex-Q nebulizer system. ApexQ and ApexQ2 represent

the APEX-Q nebulizer system used with the former and the latter nebulizers, respectively. (b) Repeated measurements of

efficiency of U5000AT nebulizer system for a flow rate of 0.19 mL min[-1].

**Data availability**
The data used in this study will be submitted to the Arctic Data Archive System when the manuscript has been published.

**Author contributions**
KGA designed the study and led the manuscript writing. RD, MH, KGA, and KeK built the CFA system at NIPR. NM, TM,
SO, YK, and MK developed the improved method for rBC analyses, including the calibration method. YOT, RD, JO, and KyK
performed the CFA analyses of the SIGMA-D core. YOT measured nebulizer efficiencies and performed rBC loss tests. YOT,
JO and MH performed dispersion tests. SM, KoF, NN, and AT dated the core. KGA, YOT, and KaF analysed the CFA data.
MH and SM measured ion concentrations. TA designed and led the ice coring project at SIGMA-D. All the authors discussed
the results.

**Competing interests**
The authors declare that they have no conflict of interest.

**Acknowledgements**
We would like to thank Hideaki Motoyama for drilling the SIGMA-D core and Yuki Komuro for cutting and processing the
core in the field. We also express our gratidude to Kazuhiro Hayashi for assisting with the dispersion tests. We are grateful to
the University of Copenhagen and the University of Maine for providing the melt head designs. We acknowledge Margit
Schwikowski and Joe McConnell for their valuable advice in developing the CFA-rBC system at NIPR. Additionally, we
appreciate the helpful comments from the three anonymous reviewers. This study has been supported by the JSPS KAKENHI
(Grant Numbers: JP 22221002, JP23221004, and JP18H04140), the Arctic Challenge for Sustainability (ArCS) Project
(Program Grant Number: JPMXD130000000)), the Arctic Challenge for Sustainability II (ArCS II) Project (Program Grant
Number: JPMXD1420318865), and the Environment Research and Technology Development Funds (JPMEERF20172003,
JPMEERF20202003 and JPMEERF20232001) of the Environmental Restoration and Conservation Agency of Japan.

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
