# Peer review of "Technical note: High-resolution analyses of concentrations and sizes of refractory black carbon particles deposited on northwest Greenland over the past 350 years – Part 1. Continuous flow analysis of the SIGMA-D ice core using a Wide-Range Single-Particle Soot Photometer and a high-efficiency nebulizer"

_EGUsphere, 2024_

## Author Comment (AC1)

Response to RC1

We thank Referee 1 for the very helpful and valuable comments. We will take all the comments into consideration and revise our manuscript. Our responses to the Referee's comments are shown below. The Referee's comments and our replies are numbered and shown in blue and black, respectively.

**RC1**
**General comments**

**RC1-1** The pre-print manuscript "Technical note: High-resolution analyses of concentrations and sizes of black carbon particles deposited on northwest Greenland over the past 350 years – Part 1. Continuous flow analysis of the SIGMA-D ice core using a Wide-Range Single-Particle Soot Photometer and a high-efficiency nebulizer " by Kumiko Goto-Azuma et al. describes and examines an instrumental coupling of a CFA feeding a Marin-5 nebulizer and a wide range SP2 to analyze the black carbon profile (size spectrum, number and mass) along a Greenland ice core. This article is the first part of the study, which will be supplemented by a companion paper devoted to the results of the analyses.

The article describes in great detail the analytical system used and provides a serious analysis of its performance. In several paragraphs, the authors compare their system with other existing CFA/Nebulizer/SP2 coupling, attempting to demonstrate that the latter underestimate BC masses for reasons of reduced or unstable nebulizer efficiency, or the size spectrum truncated by the classic SP2. To my knowledge, this comparison reflects parameters that are not considered by the authors, which brings into question some of their conclusions. Further investigation would be required to finalize this work.

**AC1-1** Many thanks for the general comments. We will consider all the parameters pointed out by Referee 1 in the specific comments.

**Specific comments**

**RC1-2** Lines 76-78 : While the original SP2 allows incandescence measurements of BC particles with diameters of between 70 and 850-900nm (40-450nm according to DMT), the new SP2-XR model on the market is, according to DMT, suitable for incandescence measurements of BC particles

with larger diameters ranging from 50 to 800nm. To date, I know of no research team working on snow and ice samples with this new SP2-XR, but it is used for atmospheric measurements.

You will also note the difference in size range between the data proposed by DMT and that of (Mori et al, 2016) for the original SP2 indicating a factor of 2 in measurable sizes. A specific study as done by (Mori et al, 2016) with an SP2-XR would suggest that the SP2-XR would be able to cover a wider range up to >1.5μm without modification. Unfortunately, this study is not available.

**AC1-2** Thank you very much for the important comments on off-the-shelf SP2s provided by DMT. There was confusion about the measurable size ranges for different versions of off-the-shelf SP2s and the SP2 with the modifications made by Moteki et al. (2010). The standard SP2 referred to by Mori et al. (2016) meant a version with the modifications made by Moteki et al. (2010), which is not the off-the-shelf SP2 or SP2-XR. We mistakenly compared Moteki et al.'s (2010) version of SP2 with the Wide-Range SP2 developed by Mori et al. (2016). We will correct the mistake and compare the Wide-Range SP2/Marin 5 setup with the off-the-shelf classic SP2 (measurable size range presented in the brochure was 70-500 nm) / U5000AT setup and the off-the-shelf SP2/APEX-Q setup.

We are not convinced by the operator manual of the SP2-XR that it can cover a wider size range up to >1.5 μm. If the study is not available (has not been published), we are unable to refer to this study in the manuscript.

**RC1-3** Line 117-118 : Can you explain how you obtained this depth resolution value of 0.3±0.1mm ? Is it only the resolution related to the laser positioning sensor or the resolution once the water is analyzed by the different online instruments ? Is this value defined solely by the fluidics of the CFA system (line, valves, debubbler, …) ?

**AC1-3** The depth resolution mentioned is that of the laser positioning sensor, not the resolution of the water analysis performed by different online instruments. We apologize for any confusion caused by the text. We will revise it to ensure clarity. The depth resolution of the laser positioning sensor has been defined and published by Dallmayr et al. (2016). Therefore, we will not explain its definition again.

**RC1-4** Line 135 : This is not the focus of this article but can you clarify why your Picarro calibration is only done after the CFA session and not also before the measurements? Have you observed sufficient stability of the instrument if it is running all the time?

**AC1-4** After receiving this comment, we checked the log data of the CFA sessions again. Although we currently calibrate our Picarro before and after a CFA session, we calibrated it only after a CFA session for the SIGMA-D, most likely to save time. Since we ran Milli-Q water before and after every CFA session, we could confirm that our Picarro was stable enough during the 4-5 hours of each CFA session. Figure 5 indicates that the stability of the Picarro was sufficient for our purposes.

In addition to confirming the Picarro's stability, we found an error in the description of our Picarro. Although we wrote that we had used the Picarro L2130-i, we had also used the L2120-i when the L2130-i was out of order. We will correct the description of the Picarro.

**RC1-5** Line 151 : This comment concerns the supply of melt water to the nebulizer and then to the SP2. For accurate measurement of BC, the water flow must be controlled as it is proportional to the BC measurement. Peristaltic pumps are not the most stable over time and the flow rate varies according to wear on the tygon tube. What's more, this type of pump does not produce a stable flow, but a pulsed flow, as can be seen on an APEX/SP2 setup. Do you have any clarification on these points, and don't you think it's necessary to add a precise flow measurement before introduction into the nebulizer, using a Sensirion micro flow sensor for example ?

**AC1-5** Before and after each CFA session, we measured the flow rate. After each CFA session, the flow rate usually decreased slightly (~5%), likely due to wear of the tube. Just before the next CFA session, the flow rate of the peristaltic pump was adjusted. In this way, we maintained an almost constant flow rate with a variability of less than 5%. Strictly speaking, peristaltic pumps produce pulsed flow, as Referee 1 commented. However, the peristaltic pump was run at ~7.50 rpm, which was high enough that the pulses were not observed in our BC data. We will add this information to the manuscript.

**RC1-6** Line 166 + : While the internal mass and size calibration of the SP2 is relatively stable with time if the instrument is not moved, experience has shown that the nebulizer's efficiency is less so. Your method consists of measuring once this efficiency using PSL particles and Aquablack according to the size range. However, whether feeding an ICPMS or, in this case, an SP2, regular

calibration of the nebulizer is necessary and this is generally done by analyzing a range of calibration solutions of known concentration on a daily frequency. How can you demonstrate that the Marin-5 model is more stable than the other nebulizers used for this type of experiment ?

**AC1-6** We have not tested the stability of APEX-Q. Hence, it is difficult to compare the stability of APEX-Q with that of Marin-5. The repeated measurements of the nebulizer efficiency of Marin-5 (please see Fig. S1) show almost the same nebulizer efficiencies over time, though there is some variability in the nebulizer efficiency data shown in Fig. 2(a). For BC diameters < 2 μm, the error was ±8 %, which does not significantly affect the BC data. Therefore, we used the same nebulizer efficiency values. We also frequently checked the SP2/nebulizer system using PSL and confirmed the stability of the system. We will replace Fig. 2(a) with Fig. S1, or add Fig. S1 as supplementary material, and include a few lines in the text to explain the stability of Marin-5 over time. We have also compared the efficiencies of APEX-Q and Marin-5 nebulizers. We will add this comparison to the manuscript.

[Figure]

Fig. S1

**RC1-7** In addition to these technical comments, can you provide more information about the post-processing of SP2 data in order to obtain BC mass and size profiles? The SP2 data files are relatively heavy, so some users try to extract them directly from the DMT software, but others turn to the PSI ToolKit.

**AC1-7** We used the "Standard SP2 Software" and the "Probe Analysis Package for Igor (PAPI)", both provided by DMT, to acquire and process the incandescent signal in binary data and convert it to text format. Then we used our original code to calculate the mass and size of BC particles. We will add this information to the text. We did not use the PSI ToolKit.

**RC1-7** Line 243 : Figure 2 does not convince me about the stability of the nebulizer, whatever the flow rate or particle size range. For 0.384mL.min-1, for example, for the <2µm section, the efficiency varies from around 27% to 42% (Fig. 2a).

**AC1-8** We determined the nebulizer efficiency to be 34.2% ± 8.0% for BC particles < 2 µm. As Referee 1 commented, the nebulizer efficiency does vary between 27% and 42%. Therefore, when we estimated the total error in BC data, we took this variation into account following Mori et al. (2016). However, as described in AC1-6 and Fig. S1, the nebulizer efficiency did not change over long periods.

**RC1-9** Line 250 + : Signal dispersion. Dispersion tests are carried out using two solutions with different characteristics (in BC, ionic charge and isotopic composition) injected alternately through a valve under the melting head and then circulated to the analytical instruments. This is a good method, but a step is missing to estimate the impact of the melting head on this dispersion. Several parameters are not taken into account. 1) Even if the stratigraphy in the ice samples were perfectly horizontal, mixing would occur between the samples in the center of the ice stick and those on the outside of the inner ring (13mm?) mixed up to the port of the CFA line, 2) the ice strata are not always horizontal in the stick.

**AC1-9** We injected the solutions near the center hole of the melt heads (i.e., from above the melt head), not through a valve under the melt head. Since this was not clear in the manuscript, we will revise Line 204. Mixing occurs between the samples in the center of the ice stick and those on the outside of the inner wall (26 x 26 mm square-shaped melt head as described by Bigler et al. (2011)). However, due to the very short distance and very small dead volume within the melt heads, the

mixing that occurs within the melt heads is negligibly small compared to the mixing that occurs in other parts of the CFA system, such as the debubbler, valves, conductivity cells, tubing, and nebulizer.

If the stratigraphy in the ice samples is not horizontal, it does not affect the resolution of the CFA system, although it affects the temporal resolution of the ice core data. To evaluate the signal dispersion in the CFA system, we do not think that the stratigraphy in the ice samples matters. Nevertheless, the stratigraphy of the SIGMA-D core was nearly horizontal, allowing minimal mixing of ice from different ages.

**RC1-10** If we take into account only the interesting results of your method, this provides the basic parameters on the dispersion of the CFA and the analytical instruments. I'm quite surprised to see that the dispersion lengths (L1 and L2 average) are fairly similar between the instruments. It is known that the large dispersion in the Picarro is linked to a long cavity flush time, but this should be much shorter for the SP2 and ICPMS (to my knowledge closer to 10mm on other configurations). In addition, some studies have used these dispersion parameters to simulate a non-dispersed signal.

**AC1-10** The length of the tubing between the melt head and the Picarro was much shorter than the lengths of the tubing for SP2 and ICP-MS. This would explain why the dispersion lengths for the Picarro are similar to those for the SP2 and ICP-MS.

**RC1-11** L261 : Yes of course the resolution of your CFA is better than these dispersion values, you may indeed observe a signal at a higher frequency, but the values observed will be reduced by this dispersion.

**AC1-11** We completely agree with the comment. Therefore, as we wrote in the manuscript, the monthly mean values might have been affected by the preceding months.

**RC1-12** L268 + : Minimal loss of BC. That's a great information for all BC measured by CFA that should be reproduced elsewhere!

**AC1-12** Thank you for the comment. We were relieved when we saw this result.

**RC1-13** L295 + : BC profile. A rolling average over 10mm is indeed necessary to smooth out the technical characteristics of the CFA, and for an initial assessment of the data. Unfortunately, Figure 6 does not allow this work to be properly appreciated, as it is too crowded. Consideration could be given to adding an enlarged extract of the profile over a short period of a few years in order to appreciate any seasonal variation in the BC, which would be an added value to the use of the CFA and its high resolution. You can save the full profile for future publications.

**AC1-13** Apologies for Fig. 6 being too busy. We will add an enlarged extract of the profile or replace it with the current Fig. 6 as suggested by Referee 1.

**RC1-14** Lines 315 + : This brings us to the crux of the article, which proposes to demonstrate that configurations other than Nebulizer Marin-5 and WR-SP2 underestimate BC mass concentrations by XX%. It's not just the instrument and the measurement that come into play, but also the data processing. The low size limit of traditional SP2s is well known, which is also why DMT now offers an SP2-XR. Just because there are no measurements taken on sizes above 650 or 850nm does not mean that this part of the size spectrum is not considered. As shown in Figure 7, a Normalized dM/dlogD fit can be used to calculate the total mass (lognormal fit size distribution). This fit does not necessarily require measurements above 650 or 850nm to be correct if most of the peak is covered. To the best of my knowledge, but you can get in touch with the main users, to overcome the problem of the size spectrum being truncated at the top, classic SP2 users use the PSI ToolKit, which proposes the use of this fit in order to extract correct mass values. This last point should change the hasty conclusions of this manuscript.

For users of the U5000T nebulizer, on the other hand, there is a real problem of underestimation coming from instable nebulizer's efficenty.

**AC1-14** We appreciate these very important comments. Although the U5000AT ultrasonic nebulizer has recently been replaced by the APEX-Q nebulizer, many previous ice-core BC studies, including most of those in Greenland, used the U5000AT nebulizer. To investigate spatial variability within Greenland and the Arctic, we need to compare our new data with the valuable BC data previously obtained with the U5000AT nebulizer. Thus, we think it is important to evaluate the degree of underestimation of the BC data obtained with the classic SP2/U5000AT nebulizer setup. When we compare the two setups, we will cut off the BC particles > 500 nm (not 650 or 850 nm as written in

the current version of our manuscript) because the measurable size range presented in the brochure of a classic SP2 was 70-500 nm.

We also think it is important to compare the Wide-Range SP2/Marin-5 setup with the classic SP2/APEX-Q setup, which is used in more recent ice-core BC studies. We agree that the total mass concentration of BC can be more accurately estimated assuming a lognormal size distribution if most of the peak is covered. We will investigate recent studies using the classic SP2/APEX-Q setup and revise this manuscript accordingly. However, we would like to point out that the size distribution of BC in snow or ice cores does not always follow a lognormal distribution. Bimodal size distributions with second peaks > 500 nm have been reported by Mori et al. (2019) and Kinase et al. (2020). Unless the size distribution is actually measured with a Wide-Range SP2, we cannot ensure that the size distribution follows a lognormal distribution and that the total mass can be accurately calculated assuming the lognormal distribution. We will revise the manuscript to include these issues.

**Technical corrections**

As a non-native English speaker, I will not be making any technical corrections to this manuscript.

---

## Author Comment (AC2)

Response to RC2

We thank Referee 2 for the very valuable and helpful comments. We would like to revise the manuscript, taking all the comments into consideration. Our responses to the Referee's comments are shown below. The Referee's comments and our replies are numbered and shown in blue and black, respectively.

**RC2**

**RC2-1** This manuscript by Goto-Azuma et al. describes a continuous flow ice core analysis system with parameters used for the analysis of the SIGMA-D core from NW Greenland. This manuscript aims to describe the NIPR CFA system (including SP2, ICPMS, Picarro, etc.), conduct a detailed assessment of continuous ice-core BC analysis with the Marin 5 and wide range SP2 system, and introduce the analysis and dating of the SIGMA-D ice core. My overall impression of this manuscript is that while the methods presented here underpin some very interesting BC data from the SIGMA-D core (which are presented in a companion paper), it does not have sufficient novelty or focus to stand as a separate manuscript. I would suggest the authors revisit the purpose of this manuscript and reframe it with a more central goal as I think most of what is included would be more appropriate for the methods section of the science-focused manuscript. Hopefully my suggestions below are useful. I do think the resulting datasets (discussed in the companion paper) are very interesting and appear to be quite robust, but reiterate that I do not think this methods manuscript holds up very well on its own in its current form.

**AC2-1** Although the BC measurement technique using the Wide-Range SP2 and the Marin-5 nebulizer has already been reported by Mori et al. (2016), this manuscript presents the first attempt to apply this method to a CFA system, allowing continuous and high-resolution measurements of the size distribution as well as concentrations of BC particles in ice cores. We believe it is important to describe such a combined system and assess its performance. Furthermore, to utilize and maximize the valuable data obtained by pioneering work such as that of McConnell et al. (2007), which used a classic SP2 and ultrasonic nebulizer U5000AT, we need to estimate the degree of underestimation for such a classic system. However, some important information and details are missing from the current manuscript, as pointed out by Referees 1 and 2. If we make revisions following all the Referees' comments and add the necessary technical information, we believe that a revised manuscript could have sufficient novelty on its own.

As for the general description of the entire CFA system and the units other than the BC unit, we could simplify or move some parts to the Appendix or Supplementary Material, as suggested by Referee 3. It is not new to use SP2, ICP-MS, Picarro, etc., with a melting system. Nevertheless, our CFA system enabled simultaneous analyses of many parameters in one laboratory, which we believe is unique. Although multi-parameter CFA analyses of ice cores have been previously conducted during CFA campaigns using different measurement units, those units were usually brought to a CFA laboratory by multiple laboratories and used only during the campaign. To our knowledge, CFA systems used in such campaigns using different units have been rarely reported. We think it is worth introducing at least briefly the general features of the CFA system built at the National Institute of Polar Research, which covers a wide range of analyses and consistently analyzes all the parameters. We also think it is worth briefly assessing the performance of the different units in the CFA system.

**RC2-2** First, the title and abstract indicate the main goal of this manuscript is to present the application of the wide-range SP2 + Marin 5 for continuous ice core analysis. The SP2+nebulizer system has been used for continuous analysis in a number of ice core labs and the details of this specific system for BC measurements in liquid water have been presented previously (Mori at al., 2016), so the assessment of the modified SP2 and Marin 5 nebulizer system is not particularly novel. Most of the other methods presented here (e.g. ice core CFA SP2, Picarro, and ICPMS analysis) are also well-established, with the exception of the BC particle size measurements, and therefore are more appropriate for a methods section of a science-oriented paper in my opinion.

**AC2-2** We believe that the assessment of the modified SP2/Marin 5 nebulizer system attached to a melting system and the evaluation of underestimation by previous BC measurements have sufficient novelty for the following two reasons. First, to our knowledge, the dispersion of BC particles, potential losses of BC particles, and resulting changes in BC size distribution in a CFA system have never been assessed. Second, the degree of underestimation of BC mass concentration for the classic SP2/nebulizer system has not been quantitatively evaluated. We would like to emphasize these points in our revised manuscript. Including these results in the methods section of a science-oriented manuscript (Part 2 of our study) would obscure the purpose of the science-oriented manuscript and would be distracting. Moreover, it would be too lengthy. We believe that the important information derived from the Part 1 manuscript should be highlighted; hence, it is not appropriate for Supplementary Material. Therefore, we would like to separate the methods paper (Part 1 of our study) from the science-oriented paper (Part 2 of our study), subject to approval by the Referees and

Editor. We admit that the current manuscript may give the impression that it lacks sufficient novelty. We plan to add more technical information and make revisions to address Referee 2's concerns.

**RC2-3** The measurement of BC size distributions throughout the core, though, is quite novel and exciting, but this manuscript lacks detail or justification for this specific measurement. There have not been published long-term reconstructions of BC particles size from Arctic ice cores, largely because as the authors correctly state, "obtaining accurate estimation of the size distribution of BC particles on a routine basis is not easy" (line 67). I agree, and a major reason why is because it is extremely difficult to maintain a stable SP2 response/calibration and nebulizer efficiency throughout an ice core CFA campaign. However, the manuscript did not justify how the authors have overcome these challenges to apply this method to ice core CFA analysis, where it is crucial to demonstrate stability and reproducibility of the method to ensure consistent measurements over weeks and/or months of ongoing analysis. I think more detail is warranted on how the authors ensure a coherent BC size dataset throughout the SIGMA-D analysis, which likely spanned a few months given the stated analysis rate of 6-7 m on one to two analysis days per week. Was the SP2-Marin5 system stable throughout an analysis day, week, month, etc.? Were replicate ice sections analyzed with good agreement? What protocols were used or standards run to ensure a consistent dataset? How was SP2 data processed? Investigating these questions will require presenting some timeseries BC size distribution data, which is omitted entirely in this manuscript despite its emphasis in the title and abstract. Only BC mass and number concentration timeseries are shown and even then, the figures are too small to evaluate the timeseries data.

**AC2-3** We agree with these comments, which were also stated by Referee 1. We apologize for not sufficiently describing the stability of the SP2/Marin 5 nebulizer system. We will demonstrate the stability and reproducibility of our method to ensure consistent measurements over a long period of time, as noted in our reply (AC1-6) to Referee 1. Additionally, the stability of the system was ensured by repeated measurements of the same samples over several months or a couple of years, as reported by Mori et al. (2019). We will also briefly explain how we processed the data, as written in our reply to Referee 1 (AC1-7). Unfortunately, we could not analyze replicate ice sections for BC due to the limited amount of the SIGMA-D core. Instead, we believe that the results of the BC loss test presented in Fig. 4 ensure the reproducibility of our method. Again, we apologize for the overly busy Figures 6 and 8. We will add enlarged extracts of the profiles or replace the current figures with them. We will also consider presenting more of the BC size data in a way that does not overlap with the Part 2 manuscript of this study.

**RC2-4** Other aspects of the BC dataset that would be valuable to assess would be the Marin 5's performance against the Apex Q, which is more prevalent now for ice/snow analysis than the Cetac U5000AT and also has much better nebulization efficiency for large particles (Wendl et al., 2014). While the Cetac was originally the nebulizer of choice for the SP2 ice core method when it was first introduced (McConnell et al., 2007), I don't think the Cetac should be the benchmark for the underestimation of BC concentration for a 'standard' ice core method anymore since many groups have moved away from it (largely because of its efficiency and stability issues). Lastly, it should be made explicit that many of the findings related to BC concentration underestimation in ice cores presented here apply primarily to Arctic and alpine ice core sites. The choice of nebulizer (Cetac U500AT vs Apex Q, at least) does not seem to impact BC concentrations for Antarctic ice cores sites as much given the much smaller particles and lower BC concs observed at those sites (Arienzo et al., 2016, JGR, Supplemental Fig 1).

**AC2-4** We appreciate the constructive suggestion to compare the performances of the Marin-5 and APEX-Q nebulizers. We did compare the efficiencies of both nebulizers for the size range between ~200 and ~700 nm at a flow rate of 0.384 mL/min (Fig. S2). We will add Fig. S2 in the text or as supplementary material. Fig. S2 shows that the efficiency of the Marin-5 is slightly higher than that

[Figure]

Fig. S2

of the APEX-Q. However, we could not perform stability tests for the APEX-Q or analyze the nebulizer efficiency for larger BC particles because we did not have an APEX-Q in our institute. We could borrow it from a distributor only for a short period.

We could not find Arienzo et al., 2016 in JGR. Do you mean Arienzo et al., 2017, JGR? If you mean Arienzo et al., 2017, JGR, Supplemental Fig. 1 presents concentrations only. We could not find any size distribution data. Could you please provide more information on the paper so that we can download it?

We believe that many of the findings related to BC concentration underestimation in ice cores apply not only to Arctic and alpine ice core sites, but also to Antarctic sites. The size distributions of BC in the surface snow of Eastern Antarctica reported by Kinase et al. (2020, JGR) indicated that the mass ratios of BC particles > 500 nm were large, although the concentrations were very low.

**RC2-5** Other sections of the manuscript, including the description of the complete CFA setup with the new addition of the ICPMS and preliminary dating of the SIGMA-D ice core, seemed extraneous and distracting to me from the more exciting BC size distribution idea. As I mentioned previously, I think those sections are more appropriate for the methods section of the science focused manuscript as they are largely established methods. Additionally, the dating section did not include enough detail to be compelling (for example the dating section only showed ~3 m of annual layer counting and did not show the tritium ties or volcanic synchronization).

**AC2-5** As we wrote earlier in our reply (AC-2-1), we will simplify or move some parts to supplementary material so that the manuscript does not appear extraneous or distracting. However, we would like to separate Part 1 and Part 2 as explained earlier. Since the dating method, including tritium and sulfate data, has already been published elsewhere (Nagatsuka et al., 2021), we did not repeat it in this manuscript. We will consider moving the dating section to the Part 2 manuscript.

**RC2-6** In short, this Part 1 manuscript, which is framed as a BC methods paper by the title and abstract, does not have sufficient novelty or detail to stand alone in its current form. In my opinion, it is better suited to be included as a methods section for the scientific paper unless the manuscript is refocused around the novel BC size distribution method.

**AC2-6** As we wrote earlier, we would like to make revisions to the manuscript, emphasizing the novel method for size distribution measurements, to ensure it has sufficient novelty.

Other comments

**RC2-7** Line 117: 0.3 +/- 0.1 mm depth resolution seems incorrect- are the units right?

**AC2-7** Yes, the units are correct. As mentioned in our reply to Referee 1 (AC1-3), this depth resolution is that of the laser positioning sensor, not the resolution once the water is analyzed by the different online instruments. We apologize for the confusing text. We will revise it to avoid any confusion. The depth resolution of the laser positioning sensor has been defined and published by Dallmayr et al. (2016).

**RC2-8** Line 288-289: If the dating section stays, it would be worth including a figure showing volcanic synchronization. What is meant by 'made adjustment' prior to 1783? What exactly was adjusted?

**AC2-8** As we mentioned earlier (AC2-5), the results of the volcanic synchronization have been presented in a previous paper by Nagatsuka et al. (2021). Therefore, we do not want to repeat it in this manuscript. However, if requested by the Referees or Editor, we can add a figure showing volcanic synchronization. In any case, we will indicate which volcanic peaks were used to refine the previous dating results. Having said this, we will also consider moving the entire dating section to the Part 2 manuscript of our study if it is deemed more appropriate.

**RC2-9** Lines 298-300: Are the sporadic peaks attributed to large particles reproducible? What do the BC size distributions look like for those depths? It would be interesting to understand if any meaningful interpretations can be drawn from them. If they are just filtered out of the data and considered noise, then what is there any advantage of using the wide-range SP2 over a standard one?

**AC2-9** Thank you very much for the important comments. Many of the sporadic peaks are attributed to large particles, which are not always reproducible. However, large BC peaks are often found in other ice cores from Greenland and in the SIGMA-D $NH_4^+$ record. Such peaks are likely due to large boreal forest fires, as presented in the Part 2 manuscript of our study. At these peaks, BC sizes are larger. When we revise the manuscript, we will show size distribution data for such BC peaks.

Although many of the sporadic peaks are filtered out of the data, even 10 mm averages of the raw data show high concentration peaks. The data averaged over 10 mm (Fig. 8) show a difference in mass concentrations for different upper limits of the measurable BC size. However, Figure 8 is too busy, as commented by Referees 1 and 2, and this feature cannot be seen very well. We will present an enlarged extract of the profile to show this feature more clearly. By improving Figure 8, we hope the advantage of using the Wide-Range SP2 over a standard one will be clearer. Figure 9 also shows the advantage of using the Wide-Range SP2 over a standard one.

**RC2-10** Lines 329-331: While the CFA system is capable of measuring water isotopes, ICPMS, microparticles, and methane, I don't think that is demonstrated in this manuscript and distracts from the BC focus.

**AC2-10** We will revise the manuscript as stated earlier to address this comment.

**RC2-11** Does the paper address relevant scientific questions within the scope of ACP?

Yes, ice core BC size distribution and concentration measurements are within the scope of ACP.

**RC2-12** Does the paper present novel concepts, ideas, tools, or data?

The BC size distribution method is novel, but the other aspects of the manuscript (dating, ice core CFA analysis) not so much.

**AC2-12** To address this comment, we will revise the manuscript as stated earlier.

**RC2-13** Are substantial conclusions reached?

No. I do not think the manuscript reaches substantial conclusions, as the novel aspect of the manuscript (the BC particle mass method) is not well described, and the results/conclusions of the SIGMA-D analysis are discussed in a companion paper.

**AC2-13** To address this comment, we will revise the manuscript as stated earlier.

**RC2-14** Are the scientific methods and assumptions valid and clearly outlined? Are the results sufficient to support the interpretations and conclusions? Is the description of experiments and calculations sufficiently complete and precise to allow their reproduction by fellow scientists (traceability of results)?

No. More information needed on the stability and reproducibility of the BC size distribution method over the course of the ice core analysis. The BC size distribution records are not presented in this manuscript making it difficult to assess the method.

**AC2-14** We will add more information on the stability and reproducibility of the BC size distribution method, as stated earlier. We will also consider presenting the size distribution data.

**RC2-15** Do the authors give proper credit to related work and clearly indicate their own new/original contribution?

Yes

**RC2-16** Does the title clearly reflect the contents of the paper? Does the abstract provide a concise and complete summary? Is the overall presentation well structured and clear?

Somewhat. The title and abstract focus on BC measurements, but the manuscript also includes sections about the full NIPR CFA system and SIGMA-D ice core dating that I found distracting.

**AC2-16** We will restructure the distracting parts of the manuscript.

**RC-17** Is the language fluent and precise? Are mathematical formulae, symbols, abbreviations, and units correctly defined and used?

Yes

**RC-18** Should any parts of the paper (text, formulae, figures, tables) be clarified, reduced, combined, or eliminated?

Yes. Described in comments above.

**AC2-18** We would like to keep this manuscript separate from Part 2 manuscript of our study for the reasons stated above.

**RC2-19** Are the number and quality of references appropriate?

Yes, though lacks citations to more recently published Arctic BC records.

**AC2-19** We will add more recently published Arctic BC records to the references.

---

## Author Comment (AC3)

**Response to RC3**

We thank Referee 3 for the very helpful and valuable comments. We will take all the comments into consideration and revise our manuscript. Our responses to the Referee's comments are shown below. The Referee's comments and our replies are numbered and shown in blue and black, respectively.

This manuscript "Technical note: High-resolution analyses of concentrations and sizes of black carbon particles deposited on northwest Greenland over the past 350 years – Part 1. Continuous flow analysis of the SIGMA-D ice core using a Wide-Range Single-Particle Soot Photometer and a high-efficiency nebulizer" submitted by Goto-Azuma et al. describes an improved CFA system by coupling single-particle soot photometer and a high-efficiency nebulizer. This technique is suitable to perform high-resolution measurements of black carbon (BC) regarding concentration, as well as size distribution up to 4 μm. The authors applied this technique to analyze the BC particles in an ice core retrieved at the SIGMA-D site from the northwest Greenland. This manuscript is accompanied by a following part focusing on the 350-year BC record of the SIGMA-D ice core. This work has advanced the conventional CFA system, especially concerning to the size distribution of BC that has been less considered before. Therefore, this specific merit deserves a publication in an esteemed journal such as ACP.

**RC3-1** I suggest that the authors might discuss the innovative content (e.g., size distribution of BC particles) in more details, and simply the other parts that have been considered thoroughly in previous papers, or to include the other parts in the supplementary material. Accordingly, the introduction should be revised for a concise review of previous works, but focus more on its novelties.

**AC3-1** Thank you for the constructive comments. We will revise the manuscript to address them accordingly.

**Other comments:**

**RC3-2** Lines 304-314: The authors claim that a combination of the standard SP2 and a high efficiency nebulizer, and a combination of the standard SP2 and a traditional ultrasonic nebulizer would lead to underestimation of the averaged mass concentration by 12% and 17%, respectively.

However, I don't know if the authors analyzed BC concentration using these two conventional methods. Please provide more details to reach this conclusion.

**AC3-2** Our apologies for the confusing explanation. We did not analyze the BC concentration using conventional methods. Instead, we estimated the underestimations based on the measurable size ranges for conventional methods. We will revise the manuscript to make this clear.

**RC3-3** Line 29: Please take a check on the resolution value of 10-40 mm.

**AC3-3** The resolution values depend on the definition of 'resolution.' We presented two types of resolutions, 10 mm and 40 mm. However, this was not clear in the abstract. Since we cannot explain the details in the abstract, we will delete '(resolution: 10-40 mm)' from the abstract to avoid any confusion. Instead, we will explain this more clearly in the text.

**RC3-4** Line 58:   The words "have become possible" should be deleted.

**AC3-4** Thank you for pointing out the typo. We will delete the words.

**RC3-5** Lines 117-118: What's the meaning for saying the depth resolution value of $0.3 \pm 0.1$ mm?

**AC3-5** This is the depth resolution of the laser positioning sensor, which has been published by Dallmayr et al. (2016). However, as pointed out by Referees 1 and 3, this sentence was very confusing. We will revise the manuscript to avoid confusion.

**RC3-6** Lines 186-187: Please explain in more details how to calculate the reproducibility.

**AC3-6** We repeated measurements of the same samples. We will revise the manuscript and explain how we calculated the reproducibility.

**RC3-7** Line 224: "$\pm 0:05‰$"→"$\pm 0.05‰$"

**AC3-7** We were not aware of this typo. Thank you for pointing it out. We will correct it.

**RC3-8** Line 238: "± 0:08‰"→"± 0.08‰"

**AC3-8** We were not aware of this typo. Thank you for pointing it out. We will correct it.

---

## Author Response (AR1)

We thank Referees 1, 2, and 3 for their thorough reviews and insightful comments, which have significantly improved the manuscript. Our responses to the Referee's comments are shown below. The Referee's comments and our replies are numbered and shown in blue and black, respectively.

**Response to RC1**

We thank Referee 1 for the very helpful and valuable comments. We have taken all the comments into consideration and revised our manuscript. Our responses to the Referee's comments are shown below.

**RC1**

**General comments**

**RC1-1** The pre-print manuscript "Technical note: High-resolution analyses of concentrations and sizes of black carbon particles deposited on northwest Greenland over the past 350 years – Part 1. Continuous flow analysis of the SIGMA-D ice core using a Wide-Range Single-Particle Soot Photometer and a high-efficiency nebulizer " by Kumiko Goto-Azuma et al. describes and examines an instrumental coupling of a CFA feeding a Marin-5 nebulizer and a wide range SP2 to analyze the black carbon profile (size spectrum, number and mass) along a Greenland ice core. This article is the first part of the study, which will be supplemented by a companion paper devoted to the results of the analyses.

The article describes in great detail the analytical system used and provides a serious analysis of its performance. In several paragraphs, the authors compare their system with other existing CFA/Nebulizer/SP2 coupling, attempting to demonstrate that the latter underestimate BC masses for reasons of reduced or unstable nebulizer efficiency, or the size spectrum truncated by the classic SP2. To my knowledge, this comparison reflects parameters that are not considered by the authors, which brings into question some of their conclusions. Further investigation would be required to finalize this work.

**AC1-1** Many thanks for the very important comments. We have considered all the parameters pointed out by Referee 1 in the specific comments.

**Specific comments**

**RC1-2** Lines 76-78 : While the original SP2 allows incandescence measurements of BC particles with diameters of between 70 and 850-900nm (40-450nm according to DMT), the new SP2-XR model on the market is, according to DMT, suitable for incandescence measurements of BC particles with larger diameters ranging from 50 to 800nm. To date, I know of no research team working on snow and ice samples with this new SP2-XR, but it is used for atmospheric measurements.

You will also note the difference in size range between the data proposed by DMT and that of (Mori et al, 2016) for the original SP2 indicating a factor of 2 in measurable sizes. A specific study as done by (Mori et al, 2016) with an SP2-XR would suggest that the SP2-XR would be able to cover a wider range up to >1.5µm without modification. Unfortunately, this study is not available.

**AC1-2** Thank you very much for the very important comments on off-the-shelf SP2s provided by DMT. Without RC1-2, we would not have noticed our serious mistake. There had been confusion about the measurable size ranges for different versions of off-the-shelf SP2s and the SP2 with the modifications made by Moteki and Kondo (2010). The "standard SP2" referred to by Mori et al. (2016) meant a version with the modifications made by Moteki and Kondo (2010), which was not the off-the-shelf SP2 or SP2-XR. We had mistakenly compared Moteki et al.'s (2010) version of SP2 with the Wide-Range SP2 developed by Mori et al. (2016). In the revised manuscript, we corrected the mistake and compared (1) the Wide-Range SP2, (2) the SP2 modified by Moteki and Kondo (2010) or off-the-shelf SP2-XR, and (3) the off-the-shelf classic SP2 (measurable size range presented in the brochure was 70-500 nm). For this comparison, we assumed to use a nebulizer with high and size independent efficiency as Marin-5 and APEX-Q.

We are not convinced by the operator manual of the SP2-XR that it can cover a wider size range up to >1.5 µm. If the study is not available (has not been published), we are unable to refer to this study in the manuscript.

**RC1-3** Line 117-118 : Can you explain how you obtained this depth resolution value of 0.3±0.1mm ? Is it only the resolution related to the laser positioning sensor or the resolution once the water is analyzed by the different online instruments ? Is this value defined solely by the fluidics of the CFA system (line, valves, debubbler, …) ?

**AC1-3** The depth resolution mentioned is that of the laser positioning sensor, not the resolution of the water analysis performed by different online instruments. We apologize for any confusion caused by the text. We have revised it to ensure clarity. The depth resolution of the laser positioning sensor has been defined and published by Dallmayr et al. (2016). Therefore, we did not explain its

definition again. The explanation regarding the laser positioning sensor has been moved to Appendix A in the revised manuscript to address Referee 2's comment (RC2-5).

**RC1-4** Line 135 : This is not the focus of this article but can you clarify why your Picarro calibration is only done after the CFA session and not also before the measurements? Have you observed sufficient stability of the instrument if it is running all the time?

**AC1-4** After receiving this comment, we checked the log data of the CFA sessions again. Although we currently calibrate our Picarro both before and after a CFA session, we calibrated it only after a CFA session for the SIGMA-D, most likely to save time. However, we ran Milli-Q water before and after every CFA session, which allowed us to confirm that our Picarros remained stable during the 4-5 hours of each CFA session. The good agreement between the CFA data (from Section A of the SIGMA-D core) and discrete sample data (from Section B of the core), though not shown in the revised manuscript to address Referee 2's comments, confirmed that that the Picarro's stability was sufficient for our purposes. In Appendix A, we have added a brief explanation regarding the Picarro's stability.

In addition to confirming the Picarro's stability, we found an error in the description of our Picarro. Although we wrote that we had used the Picarro L2130-i, we had also used the L2120-i when the L2130-i was out of order. We have corrected the description of the water isotope unit.

**RC1-5** Line 151 : This comment concerns the supply of melt water to the nebulizer and then to the SP2. For accurate measurement of BC, the water flow must be controlled as it is proportional to the BC measurement. Peristaltic pumps are not the most stable over time and the flow rate varies according to wear on the tygon tube. What's more, this type of pump does not produce a stable flow, but a pulsed flow, as can be seen on an APEX/SP2 setup. Do you have any clarification on these points, and don't you think it's necessary to add a precise flow measurement before introduction into the nebulizer, using a Sensirion micro flow sensor for example ?

**AC1-5** Before and after each CFA session, we measured the flow rate. After each CFA session, the flow rate usually decreased slightly (~5%), likely due to wear of the tube. Just before the next CFA session, the flow rate of the peristaltic pump was adjusted. In this way, we maintained an almost constant flow rate with a variability of less than 5%. Strictly speaking, peristaltic pumps produce pulsed flow, as Referee 1 commented. However, the peristaltic pump was run at ~7.50 rpm, which

was high enough that the pulses were not observed in our BC data. We have added this information to the revised manuscript.

**RC1-6** Line 166 + : While the internal mass and size calibration of the SP2 is relatively stable with time if the instrument is not moved, experience has shown that the nebulizer's efficiency is less so. Your method consists of measuring once this efficiency using PSL particles and Aquablack according to the size range. However, whether feeding an ICPMS or, in this case, an SP2, regular calibration of the nebulizer is necessary and this is generally done by analyzing a range of calibration solutions of known concentration on a daily frequency. How can you demonstrate that the Marin-5 model is more stable than the other nebulizers used for this type of experiment ?

**AC1-6** Repeated measurements of the efficiency of the Marin-5 nebulizer demonstrated nearly same efficiencies over a ten-year period, though there are some fluctuations around the regression lines. For BC diameters < 2 μm, the error was ±8 %, which does not significantly affect the BC data. Therefore, we used the same nebulizer efficiency values. We also frequently checked the WR-SP2/nebulizer system using same sampes, and confirmed the stability of the system. We have added a new Figure (Fig. 3) and a few lines in the text to explain the stability of Marin-5 over time. We have also compared the efficiencies of APEX-Q, A5000UT, and Marin-5 nebulizers. We have added the results of the comparison to the manuscript (Fig.C1). However, we have not tested the stability of APEX-Q. Hence, we could not compare the stability of APEX-Q with that of Marin-5.

[Figure]

**Figure 3:** Repeated measurements of Marin-5 nebulizer efficiency over ten years.

**RC1-7** In addition to these technical comments, can you provide more information about the post-processing of SP2 data in order to obtain BC mass and size profiles? The SP2 data files are relatively heavy, so some users try to extract them directly from the DMT software, but others turn to the PSI ToolKit.

**AC1-7** We used the "Standard SP2 Software" and the "Probe Analysis Package for Igor (PAPI)", both provided by DMT, to acquire and process the incandescent signal in binary data and convert it to text format. Then we used our original code to calculate the mass and size of BC particles. We have added this information to the text. We did not use the PSI ToolKit.

**RC1-7** Line 243 : Figure 2 does not convince me about the stability of the nebulizer, whatever the flow rate or particle size range. For 0.384mL.min-1, for example, for the <2µm section, the efficiency varies from around 27% to 42% (Fig. 2a).

**AC1-8** We determined the nebulizer efficiency to be $34.2\% \pm 8.0\%$ for BC particles < 2 μm. As Referee 1 commented, the nebulizer efficiency does vary between 27% and 42%. Therefore, when we estimated the total error in BC data, we took this variation into account following Mori et al. (2016). However, as described in AC1-6 and Fig. 3, the nebulizer efficiency did not change over a ten-year period.

**RC1-9** Line 250 + : Signal dispersion. Dispersion tests are carried out using two solutions with different characteristics (in BC, ionic charge and isotopic composition) injected alternately through a valve under the melting head and then circulated to the analytical instruments. This is a good method, but a step is missing to estimate the impact of the melting head on this dispersion. Several parameters are not taken into account. 1) Even if the stratigraphy in the ice samples were perfectly horizontal, mixing would occur between the samples in the center of the ice stick and those on the outside of the inner ring (13mm?) mixed up to the port of the CFA line, 2) the ice strata are not always horizontal in the stick.

**AC1-9** We injected the solutions near the center hole of the melt heads (i.e., from above the melt head), not through a valve under the melt head. As this was not clear in the manuscript, we have added this information to the manuscript.

Mixing occurs between the samples in the center of the ice stick and those on the outside of the inner wall (26 x 26 mm square-shaped melt head as described by Bigler et al. (2011)). However, due to the very short distance and very small dead volume within the melt heads, the mixing that occurs within the melt heads is negligibly small compared to the mixing that occurs in other parts of the CFA system, such as the debubbler, valves, conductivity cells, tubing, and nebulizer.

If the stratigraphy in the ice samples is not horizontal, it does not affect the resolution of the CFA system, although it affects the temporal resolution of the ice core data. To evaluate the signal dispersion in the CFA system, we do not think that the stratigraphy in the ice samples matters. Nevertheless, the stratigraphy of the SIGMA-D core was nearly horizontal, allowing minimal mixing of ice from different ages.

We have added a few sentences to address Referee 1's concern regarding signal dispersion.

**RC1-10** If we take into account only the interesting results of your method, this provides the basic parameters on the dispersion of the CFA and the analytical instruments. I'm quite surprised to see that the dispersion lengths (L1 and L2 average) are fairly similar between the instruments. It is known that the large dispersion in the Picarro is linked to a long cavity flush time, but this should be much shorter for the SP2 and ICPMS (to my knowledge closer to 10mm on other configurations). In addition, some studies have used these dispersion parameters to simulate a non-dispersed signal.

**AC1-10** The length of the tubing between the melt head and each instrument was different. The inner diameter was also different. This would partly explain why the dispersion lengths for the Picarro are similar to those for the SP2 and ICP-MS. We have added the information on the length and the inner diameter of the tubing.

**RC1-11** L261 : Yes of course the resolution of your CFA is better than these dispersion values, you may indeed observe a signal at a higher frequency, but the values observed will be reduced by this dispersion.

**AC1-11** We completely agree with the comment. Therefore, as we wrote in the manuscript, the monthly mean values might have been affected by the preceding months. We have slightly revised the text regarding this issue.

**RC1-12** L268 + : Minimal loss of BC. That's a great information for all BC measured by CFA that should be reproduced elsewhere!

**AC1-12** Thank you for the comment. We were relieved when we saw this result.

**RC1-13** L295 + : BC profile. A rolling average over 10mm is indeed necessary to smooth out the technical characteristics of the CFA, and for an initial assessment of the data. Unfortunately, Figure 6 does not allow this work to be properly appreciated, as it is too crowded. Consideration could be given to adding an enlarged extract of the profile over a short period of a few years in order to appreciate any seasonal variation in the BC, which would be an added value to the use of the CFA and its high resolution. You can save the full profile for future publications.

**AC1-13** Apologies for Fig. 6 being too busy. We have added enlarged extracts of the profiles.

**RC1-14** Lines 315 + : This brings us to the crux of the article, which proposes to demonstrate that configurations other than Nebulizer Marin-5 and WR-SP2 underestimate BC mass concentrations by XX%. It's not just the instrument and the measurement that come into play, but also the data processing. The low size limit of traditional SP2s is well known, which is also why DMT now offers an SP2-XR. Just because there are no measurements taken on sizes above 650 or 850nm does not mean that this part of the size spectrum is not considered. As shown in Figure 7, a Normalized dM/dlogD fit can be used to calculate the total mass (lognormal fit size distribution). This fit does not necessarily require measurements above 650 or 850nm to be correct if most of the peak is covered. To the best of my knowledge, but you can get in touch with the main users, to overcome the problem of the size spectrum being truncated at the top, classic SP2 users use the PSI ToolKit, which proposes the use of this fit in order to extract correct mass values. This last point should change the hasty conclusions of this manuscript.

For users of the U5000T nebulizer, on the other hand, there is a real problem of underestimation coming from instable nebulizer's efficenty.

**AC1-14** We greatly appreciate these important comments. We had misunderstood the upper limit of the traditional off-the-shelf SP2, and as a result, made incorrect comparisons. We have now

corrected these comparisons in the revised manuscript. We have compared (1) the mass concentrations of BC particles with diameters < 4000 nm, measured using the WR-SP2 and Marin-5; (2) the mass concentrations of BC particles with diameter <500 nm, which would be measured using a traditional off-the-shelf SP2; and (3) the mass concentrations of BC particles with diameter <850 nm, which would be measured using the SP2 modified by Moteki and Kondo (2010) or the off-the-shelf SP2-XR. For (1) - (3), we have assumed a size-independent efficiency of a nebulizer, such as that of the Marin-5 and APEX-Q.

We agree that the total mass concentration of BC can be more accurately estimated by assuming a lognormal size distribution if most of the peak is covered. However, to our knowledge, most previous ice-core studies simply calculated the mass concentrations for BC particles with diameters <500 nm, (or < 620 nm range by Lim et al., (2017)). It is also important to note that the size distribution of BC in snow or ice cores does not always follow a lognormal distribution. Bimodal size distributions with second peaks > 500 nm have been reported by Mori et al. (2019) and Kinase et al. (2020). Unless the size distribution is directly measured with a Wide-Range SP2, we cannot ensure that the size distribution follows a lognormal distribution, nor can we ensure that the total mass concentration can be accurately calculated based on this assumption. We have revised the manuscript to discuss these issues.

To investigate spatial variability within Greenland and the Arctic, it is essential to compare our new data with the valuable BC data previously obtained using the off-the-shelf SP2 and the U5000AT nebulizer. However, even with the same type of the nebulizer U5000AT, there is instrumental variation (Wendl et al., 2014) and time-dependent nebulizer efficiency (Fig.C1(b)), hence, we could not estimate the nebulizer efficiency for the previously reported measurements. Consequently, we could only conclude that the underestimation would likely be greater when using the U5000AT nebulizer.

**Technical corrections**

As a non-native English speaker, I will not be making any technical corrections to this manuscript.

Response to RC2

We sincerely thank Referee 2 for the valuable and helpful comments. We have made significant revisions, carefully considering all the feedback from the Referees. While Referee 2 suggested including this manuscript (Part 1 of our study) within the methods section of the companion manuscript (Part 2), we believe that the revised manuscript presents sufficient novelty and merit to stand on its own.

**RC2**

**RC2-1** This manuscript by Goto-Azuma et al. describes a continuous flow ice core analysis system with parameters used for the analysis of the SIGMA-D core from NW Greenland. This manuscript aims to describe the NIPR CFA system (including SP2, ICPMS, Picarro, etc.), conduct a detailed assessment of continuous ice-core BC analysis with the Marin 5 and wide range SP2 system, and introduce the analysis and dating of the SIGMA-D ice core. My overall impression of this manuscript is that while the methods presented here underpin some very interesting BC data from the SIGMA-D core (which are presented in a companion paper), it does not have sufficient novelty or focus to stand as a separate manuscript. I would suggest the authors revisit the purpose of this manuscript and reframe it with a more central goal as I think most of what is included would be more appropriate for the methods section of the science-focused manuscript. Hopefully my suggestions below are useful. I do think the resulting datasets (discussed in the companion paper) are very interesting and appear to be quite robust, but reiterate that I do not think this methods manuscript holds up very well on its own in its current form.

**AC2-1** While the BC measurement technique using the Wide-Range SP2 and the Marin-5 nebulizer has already been reported by Mori et al. (2016), this manuscript presents the first application of this method to a CFA system, allowing continuous, high-resolution measurements of the size distribution as well as concentrations of BC particles in ice cores. We believe it is important to describe such a integrated system and assess its performance. Furthermore, to fully utilize and maximize the valuable data obtained previously such as the pioneering study by McConnell et al. (2007), which employed the traditional off-the-shelf SP2, it is necessary to estimate the degree of underestimation inherent in the traditional off-the-shelf SP2 which can only measure BC particles with diameters <500nm. However, as Referees 1 and 2 pointed out, some important information and details were missing from the manuscript. In response, we have made significant revisions based onl the Referees' comments and have added the necessary technical information. We believe that a revised manuscript now offers sufficient novelty to stand on its own.

As for the general description of the entire CFA system and the units other than the BC unit, we have moved some parts to Appendix A, as suggested by Referee 3. It is not new to use SP2, ICP-MS, Picarro, etc., with a melting system. Nevertheless, our CFA system enabled simultaneous analyses of many parameters in one laboratory, which we believe is unique. Although multi-parameter CFA analyses of ice cores have been previously conducted during CFA campaigns using different measurement units, those units were usually brought to a CFA laboratory by multiple laboratories and used only during the campaign. To our knowledge, assessment of the CFA systems used in such campaigns using different units have been rarely reported. We think it is worth introducing at least briefly the general features of the CFA system built at the National Institute of Polar Research, which covers a wide range of analyses and consistently analyzes all the parameters. We also think it is worth briefly assessing the performance of the different units in the CFA system.

**RC2-2** First, the title and abstract indicate the main goal of this manuscript is to present the application of the wide-range SP2 + Marin 5 for continuous ice core analysis. The SP2+nebulizer system has been used for continuous analysis in a number of ice core labs and the details of this specific system for BC measurements in liquid water have been presented previously (Mori at al., 2016), so the assessment of the modified SP2 and Marin 5 nebulizer system is not particularly novel. Most of the other methods presented here (e.g. ice core CFA SP2, Picarro, and ICPMS analysis) are also well-established, with the exception of the BC particle size measurements, and therefore are more appropriate for a methods section of a science-oriented paper in my opinion.

**AC2-2** We believe that the assessment of the modified SP2/Marin-5 nebulizer system attached to a melting system and the evaluation of underestimation by previous BC measurements have sufficient novelty for the following two reasons. First, to our knowledge, the dispersion of BC particles, potential losses of BC particles, and resulting changes in BC size distribution in a CFA system have never been assessed. Second, the degree of underestimation of BC mass concentration for the traditional off-the-shelf SP2 has not been quantitatively evaluated. We have emphasized these points in our revised manuscript. Including these results in the methods section of a science-oriented manuscript (Part 2 of our study) would obscure the purpose of the science-oriented manuscript and would be distracting. Moreover, it would be too lengthy. We believe that the important information derived from the Part 1 manuscript should be highlighted; hence, it is not appropriate for Supplementary Material. Therefore, we would like to separate the methods paper (Part 1 of our study) from the science-oriented paper (Part 2 of our study), subject to approval by the Referees and Editor. We admit that the manuscript had given the impression that it had lacked sufficient novelty.

We have added more technical information and have made revisions to address Referee 2's concerns.

**RC2-3** The measurement of BC size distributions throughout the core, though, is quite novel and exciting, but this manuscript lacks detail or justification for this specific measurement. There have not been published long-term reconstructions of BC particles size from Arctic ice cores, largely because as the authors correctly state, "obtaining accurate estimation of the size distribution of BC particles on a routine basis is not easy" (line 67). I agree, and a major reason why is because it is extremely difficult to maintain a stable SP2 response/calibration and nebulizer efficiency throughout an ice core CFA campaign. However, the manuscript did not justify how the authors have overcome these challenges to apply this method to ice core CFA analysis, where it is crucial to demonstrate stability and reproducibility of the method to ensure consistent measurements over weeks and/or months of ongoing analysis. I think more detail is warranted on how the authors ensure a coherent BC size dataset throughout the SIGMA-D analysis, which likely spanned a few months given the stated analysis rate of 6-7 m on one to two analysis days per week. Was the SP2-Marin5 system stable throughout an analysis day, week, month, etc.? Were replicate ice sections analyzed with good agreement? What protocols were used or standards run to ensure a consistent dataset? How was SP2 data processed? Investigating these questions will require presenting some timeseries BC size distribution data, which is omitted entirely in this manuscript despite its emphasis in the title and abstract. Only BC mass and number concentration timeseries are shown and even then, the figures are too small to evaluate the timeseries data.

**AC2-3** We agree with these comments, which were also stated by Referee 1. We apologize for not sufficiently describing the stability of the SP2/Marin 5 nebulizer system. We have demonstrated the stability and reproducibility of our method to ensure consistent measurements over a long period, as noted in our reply (AC1-6) to Referee 1. The stability of the system was ensured by repeated measurements of the same samples over a few year period, as reported by Mori et al. (2019). We have also briefly explained how we processed the data, as written in our reply to Referee 1 (AC1-7). Unfortunately, we could not analyze replicate ice sections for BC due to the limited amount of the SIGMA-D core. Instead, we believe that the results of the BC loss test presented in Fig. 5 ensure the reliability and reproducibility of our method. Again, we apologize for the overly busy Figures 6 and 8 (old figure numbers). We have added enlarged extracts of the profiles to Figs. 6 and 9 (Fig.6 and 8, respectively in the old version of the manuscript). We have also presented more of the BC size data including mBC data in a way that did not overlap with the Part 2 manuscript of this study.

**RC2-4** Other aspects of the BC dataset that would be valuable to assess would be the Marin 5's performance against the Apex Q, which is more prevalent now for ice/snow analysis than the Cetac U5000AT and also has much better nebulization efficiency for large particles (Wendl et al., 2014). While the Cetac was originally the nebulizer of choice for the SP2 ice core method when it was first introduced (McConnell et al., 2007), I don't think the Cetac should be the benchmark for the underestimation of BC concentration for a 'standard' ice core method anymore since many groups have moved away from it (largely because of its efficiency and stability issues). Lastly, it should be made explicit that many of the findings related to BC concentration underestimation in ice cores presented here apply primarily to Arctic and alpine ice core sites. The choice of nebulizer (Cetac U500AT vs Apex Q, at least) does not seem to impact BC concentrations for Antarctic ice cores sites as much given the much smaller particles and lower BC concs observed at those sites (Arienzo et al., 2016, JGR, Supplemental Fig 1).

**AC2-4** We appreciate the constructive suggestion to compare the performances of the Marin-5,APEX-Q, and U5000AT nebulizers. We have compared the efficiencies of these nebulizersfor (Fig. C1). We have added Fig. C1 in Appendix C. Figure C1 shows that the efficiency of the Marin-5 is slightly higher than that of the APEX-Q. However, we could not perform stability tests for the APEX-Q or analyze the nebulizer efficiency for larger BC particles because we did not have an APEX-Q in our institute. We could borrow it only for a short period.

We could not find Arienzo et al., 2016 in JGR. If it means Arienzo et al., 2017, JGR, Supplemental Fig. 1 compares the BC concentration data obtained with the APEX-Q nebulizer and those obtained with the U5000AT nebulizer, demonstrating that differences in the BC concentration between the two nebulizer systems are <5%. However, the comparison is based on the measurement of BC with diameters < 500 nm, which does not mean that the underestimation is < 5%.

We believe that many of the findings related to underestimation of BC concentration in ice cores apply not only to Arctic and alpine ice core sites, but also to Antarctic sites. The size distributions of BC in the surface snow of Eastern Antarctica reported by Kinase et al. (2020, JGR) indicated that the mass ratios of BC particles > 500 nm were large, although the concentrations were very low. We have added sentences regarding the previously reported size distributions of BC in Arctic and Antarctic snow.

[Figure]

**Figure C1: (a)** Comparison of Marin-5 and APEX-Q nebulizer efficiency for a flow rate of 0.19 mL min-1. A MicroMist U-series AR30-1-UM05E (Glass Expansion, Australia) was used for the Marin-5 nebulizer system. On the other hand, two types of nebulizers, a Conikal Nebulizer AR30-1-FC1ES (Glass Expansion, Australia) and a MicroMist U-Series nebulizer AR30-1-UM05E (Glass Expansion, Australia) were used for the Apex-Q nebulizer system. ApexQ and ApexQ2 represent the APEX-Q nebulizer system used with the former and the latter nebulizers, respectively. **(b)** Repeated measurements of efficiency of U5000AT nebulizer system for a flow rate of 0.19 mL min-1.

**RC2-5** Other sections of the manuscript, including the description of the complete CFA setup with the new addition of the ICPMS and preliminary dating of the SIGMA-D ice core, seemed extraneous and distracting to me from the more exciting BC size distribution idea. As I mentioned previously, I think those sections are more appropriate for the methods section of the science focused manuscript as they are largely established methods. Additionally, the dating section did not include enough detail to be compelling (for example the dating section only showed ~3 m of annual layer counting and did not show the tritium ties or volcanic synchronization).

**AC2-5** As mentioned earlier in our reply (AC-2-1), we have moved some parts to Appendices A and B to ensure that the manuscript remains focused and does not include extraneous or distracting content. We still believe it is reasonable to keep Part 1 and Part 2 separate, as previously explained. Additionally, we have removed the dating section including Figure 5, and plan to include it in the Part 2 manuscript.

**RC2-6** In short, this Part 1 manuscript, which is framed as a BC methods paper by the title and abstract, does not have sufficient novelty or detail to stand alone in its current form. In my opinion, it is better suited to be included as a methods section for the scientific paper unless the manuscript is refocused around the novel BC size distribution method.

**AC2-6** As we wrote earlier, we have made major revisions to the manuscript, emphasizing the novel method for size distribution measurements, to ensure it has sufficient novelty.

Other comments

**RC2-7** Line 117: 0.3 +/- 0.1 mm depth resolution seems incorrect- are the units right?

**AC2-7** Yes, the units are correct. As mentioned in our reply to Referee 1 (AC1-3), this depth resolution is that of the laser positioning sensor, not the resolution once the water is analyzed by the different online instruments. We apologize for the confusing text. We have revised it to avoid any confusion. The depth resolution of the laser positioning sensor has been defined and published by Dallmayr et al. (2016).

**RC2-8** Line 288-289: If the dating section stays, it would be worth including a figure showing volcanic synchronization. What is meant by 'made adjustment' prior to 1783? What exactly was adjusted?

**AC2-8** We have removed the entire dating section, as mentioned earlier (AC2-5). In Part 2, we will briefly explain how we dated the core. However, we will not describe the details, since major parts of the dating method has been published by Nagatsuka et al. (2023).

**RC2-9** Lines 298-300: Are the sporadic peaks attributed to large particles reproducible? What do the BC size distributions look like for those depths? It would be interesting to understand if any meaningful interpretations can be drawn from them. If they are just filtered out of the data and considered noise, then what is there any advantage of using the wide-range SP2 over a standard one?

**AC2-9** Thank you very much for the important comments. Many of the sporadic peaks are attributed to large particles, which are not always reproducible. However, large BC peaks are often found in other ice cores from Greenland and in the SIGMA-D $NH_4^+$ record. Such peaks are likely due to large

boreal forest fires, as presented in the Part 2 manuscript of our study. At these peaks, BC sizes are often larger. In the revised manuscript, we have presented examples of size distribution data for significant BC peaks.

Although many of the sporadic peaks are filtered out of the data, even 10 mm averages of the raw data show high concentration peaks. The data averaged over 10 mm (Fig. 9 in the revised manuscript, which corresponds to Fig, 8 in the submitted version of the manuscript) show a difference in mass concentrations for different upper limits of the measurable BC size. However, Figure 8 (old version) was too busy, as commented by Referees 1 and 2, and this feature cannot be seen very well. We have presented enlarged extracts of the profile to show this feature more clearly. By improving Fig. 8 (Fig. 9 in the revised manuscript), we hope the advantage of using the Wide-Range SP2 over a standard one is now clearer. Figure 10 (revised manuscript) also shows the advantage of using the Wide-Range SP2 over the off-the-shelf SP2.

**RC2-10** Lines 329-331: While the CFA system is capable of measuring water isotopes, ICPMS, microparticles, and methane, I don't think that is demonstrated in this manuscript and distracts from the BC focus.

**AC2-10** We have moved the detailed description of the CFA system to Appendix A, as stated earlier, to address this comment.

**RC2-11** Does the paper address relevant scientific questions within the scope of ACP?

Yes, ice core BC size distribution and concentration measurements are within the scope of ACP.

**RC2-12** Does the paper present novel concepts, ideas, tools, or data?

The BC size distribution method is novel, but the other aspects of the manuscript (dating, ice core CFA analysis) not so much.

**AC2-12** To address this comment, we have revised the manuscript as stated earlier.

**RC2-13** Are substantial conclusions reached?

No. I do not think the manuscript reaches substantial conclusions, as the novel aspect of the manuscript (the BC particle mass method) is not well described, and the results/conclusions of the SIGMA-D analysis are discussed in a companion paper.

**AC2-13** To address this comment, we have revised the manuscript as stated earlier.

**RC2-14** Are the scientific methods and assumptions valid and clearly outlined? Are the results sufficient to support the interpretations and conclusions? Is the description of experiments and calculations sufficiently complete and precise to allow their reproduction by fellow scientists (traceability of results)?

No. More information needed on the stability and reproducibility of the BC size distribution method over the course of the ice core analysis. The BC size distribution records are not presented in this manuscript making it difficult to assess the method.

**AC2-14** We have added more information on the stability and reproducibility of the BC size distribution method, as stated earlier. We have also added a few examples of size distribution data.

**RC2-15** Do the authors give proper credit to related work and clearly indicate their own new/original contribution?

Yes

**RC2-16** Does the title clearly reflect the contents of the paper? Does the abstract provide a concise and complete summary? Is the overall presentation well structured and clear?

Somewhat. The title and abstract focus on BC measurements, but the manuscript also includes sections about the full NIPR CFA system and SIGMA-D ice core dating that I found distracting.

**AC2-16** We have restructured the distracting parts of the manuscript.

**RC2-17** Is the language fluent and precise? Are mathematical formulae, symbols, abbreviations, and units correctly defined and used?

Yes

**RC2-18** Should any parts of the paper (text, formulae, figures, tables) be clarified, reduced, combined, or eliminated?

Yes. Described in comments above.

**AC2-18** We would like to keep this manuscript separate from Part 2 manuscript of our study for the reasons stated above.

**RC2-19** Are the number and quality of references appropriate?

Yes, though lacks citations to more recently published Arctic BC records.

**AC2-19** We have added two recently published Antarctic BC records to the references. However, we could not find any recently published Arctic ice-core data other than those already referred to in the manuscript .

**Response to RC3**

We thank Referee 3 for the very helpful and valuable comments. We have taken all the comments into consideration and have revised our manuscript. Our responses to the Referee's comments are shown below.

This manuscript "Technical note: High-resolution analyses of concentrations and sizes of black carbon particles deposited on northwest Greenland over the past 350 years – Part 1. Continuous flow analysis of the SIGMA-D ice core using a Wide-Range Single-Particle Soot Photometer and a high-efficiency nebulizer" submitted by Goto-Azuma et al. describes an improved CFA system by coupling single-particle soot photometer and a high-efficiency nebulizer. This technique is suitable to perform high-resolution measurements of black carbon (BC) regarding concentration, as well as

size distribution up to 4 μm. The authors applied this technique to analyze the BC particles in an ice core retrieved at the SIGMA-D site from the northwest Greenland. This manuscript is accompanied by a following part focusing on the 350-year BC record of the SIGMA-D ice core. This work has advanced the conventional CFA system, especially concerning to the size distribution of BC that has been less considered before. Therefore, this specific merit deserves a publication in an esteemed journal such as ACP.

**RC3-1** I suggest that the authors might discuss the innovative content (e.g., size distribution of BC particles) in more details, and simply the other parts that have been considered thoroughly in previous papers, or to include the other parts in the supplementary material. Accordingly, the introduction should be revised for a concise review of previous works, but focus more on its novelties.

**AC3-1** Thank you for the constructive comments. We have revised the introduction: we have added the importance of BC size distribution in more detail; and removed the part referring to the impacts of BC on air quality, human health etc., as this part is not relevant to Part 1, We have also moved some parts that have been considered in previous papers to Appendices.

**Other comments:**

**RC3-2** Lines 304-314: The authors claim that a combination of the standard SP2 and a high efficiency nebulizer, and a combination of the standard SP2 and a traditional ultrasonic nebulizer would lead to underestimation of the averaged mass concentration by 12% and 17%, respectively. However, I don't know if the authors analyzed BC concentration using these two conventional methods. Please provide more details to reach this conclusion.

**AC3-2** Our apologies for the confusing explanation. We did not analyze the BC concentration using conventional methods. Instead, we estimated the underestimations assuming different measurable size ranges for different versions of SP2s. We have revised the manuscript to make this clear.

**RC3-3** Line 29: Please take a check on the resolution value of 10-40 mm.

**AC3-3** The resolution values depend on the definition of 'resolution.' We presented two types of resolutions, 10 mm and 40 mm. However, this was not clear in the abstract. Since we cannot explain the details in the abstract, we have deleted '(resolution: 10-40 mm)' from the abstract to avoid any confusion. Instead, we have explained this more clearly in Section 3.2.

**RC3-4** Line 58:   The words "have become possible" should be deleted.

**AC3-4** Thank you for pointing out the typo. We have deleted the words.

**RC3-5** Lines 117-118: What's the meaning for saying the depth resolution value of $0.3 \pm 0.1$ mm?

**AC3-5** This is the depth resolution of the laser positioning sensor, which has been published by Dallmayr et al. (2016). However, as pointed out by Referees 1 and 3, this sentence was very confusing. We have revised the manuscript to avoid confusion.

**RC3-6** Lines 186-187: Please explain in more details how to calculate the reproducibility.

**AC3-6** We repeated measurements of the same samples. In the revised manuscript, we have explained how we calculated the reproducibility.

**RC3-7** Line 224: "$\pm$ 0:05‰"→"$\pm$ 0.05‰"

**AC3-7** We were not aware of this typo. Thank you for pointing it out. We have corrected it.

**RC3-8** Line 238: "$\pm$ 0:08‰"→"$\pm$ 0.08‰"

**AC3-8** This value should have been $\pm$ 0.1‰. We have corrected this error.

**Additional authors' changes in the manuscript**

1. One of the reviewers of the Part 2 manuscript requested to use the term "refractory black carbon (rBC)" instead of BC. To address this comment, we plan to revise the Part 2 manuscript. Accordingly, we have changed the word BC to rBC in the title, text, and Figures of this manuscript (Part 1).

2. We have removed Figure 5 (a figure in the submitted version of the manuscript), as we removed the whole dating section.

3. We have added Figures 3, Figures 6 (c)-(e), Figure 8, Figures 9(c) (d), and Figure C1.

4. We have modified Figures 1, 2, and 4 (only slightly).

5. We have added the number size distribution to Figure 7, which had been missing in the submitted version of the manuscript. We have also replaced the normalized mass size distribution with unnormalized one.

6. We have replaced Figure 10 (a) with a new figure, assuming 500 nm, instead of 650 nm, as the upper limit of measurable diameter of BC particles.

7. When we moved some of the methods part to Appendices A and B, we have added a few sentences, which are marked with yellow color.

8. We have revised Author contributions and Acknowledgements.